# SURF: A Simple, Universal, Robust, Fast Distribution Learning Algorithm

**Yi Hao, Ayush Jain, Alon Orlitsky, Vaishakh Ravindrakumar**
Dept. of Electrical and Computer Engineering
University of California, San Diego
{yih179, ayjain, aorlitsky, varavind}@eng.ucsd.edu

## Abstract

Sample- and computationally-efficient distribution estimation is a fundamental tenet in statistics and machine learning. We present SURF, an algorithm for approximating distributions by piecewise polynomials. SURF is: simple, replacing prior complex optimization techniques by straight-forward empirical probability approximation of each potential polynomial piece through simple empirical-probability interpolation, and using plain divide-and-conquer to merge the pieces; universal, as well-known polynomial-approximation results imply that it accurately approximates a large class of common distributions; robust to distribution mis-specification as for any degree $d \leq 8$, it estimates any distribution to an $\ell_1$ distance $< 3$ times that of the nearest degree-$d$ piecewise polynomial, improving known factor upper bounds of 3 for single polynomials and 15 for polynomials with arbitrarily many pieces; fast, using optimal sample complexity, running in near sample-linear time, and if given sorted samples it may be parallelized to run in sub-linear time. In experiments, SURF outperforms state-of-the art algorithms.

## 1 Introduction

### 1.1 Background

Estimating an unknown distribution from its samples is a fundamental statistical problem arising in many applications such as modeling language, stocks, weather, traffic patterns, and many more. It has therefore been studied for over a century, e.g. [15].

Consider an unknown univariate distribution $f$ over $\mathbb{R}$, generating $n$ samples $X^n \stackrel{\text{def}}{=} X_1, \ldots, X_n$. An *estimator* for $f$ is a mapping $\hat{f} : \mathbb{R}^n \to \mathbb{R}$. As in many of the prior works, we evaluate $\hat{f}$ using the $\ell_1$ distance, $\|\hat{f} - f\|_1$. The $\ell_1$ distance professes several desirable properties, including scale and location invariance, and provides provable guarantees on the values of Lipschitz functionals of $f$ [6].

Ideally, we would prefer an estimator that learns any distribution. However, arbitrary distributions cannot be learned with any number of samples. Let $u$ be the continuous uniform distribution over $[0, 1]$. For any number $n$ of samples, uniformly select $n^3$ points from $[0, 1]$ and let $p$ be the discrete uniform distribution over these $n^3$ points. Since with high probability collisions do not occur within samples under either distribution, $u$ and $p$ cannot be distinguished from the uniformly occurring samples. As $\|u - p\|_1 = 2$, it follows that for any estimator $\hat{f}$, $\max_{f \in \{u,p\}} \mathbb{E}\|\hat{f} - f\|_1 \gtrsim 1$.

A common modification, motivated by PAC agnostic learning, assumes that $f$ is close to a natural distribution class $\mathcal{C}$, and tries to find the distribution in $\mathcal{C}$ closest to $f$. The following notion of $\text{OPT}_{\mathcal{C}}(f)$ considers this lowest distance, and the usual *minimax learning rate* of $\mathcal{C}$, $\mathcal{R}_n(\mathcal{C})$, is the lowest worst-case expected distance achieved by any estimator,

$$\text{OPT}_{\mathcal{C}}(f) \stackrel{\text{def}}{=} \inf_{g \in \mathcal{C}} \|f - g\|_1, \quad \mathcal{R}_n(\mathcal{C}) \stackrel{\text{def}}{=} \min_{\hat{f}} \max_{f \in \mathcal{C}} \mathbb{E}_{X^n \sim f} \|\hat{f} - f\|_1.$$

As has been considered in [2], $\hat{f}$ is said to be a factor-$c$ approximation for $\mathcal{C}$ if

$$\mathbb{E}\|\hat{f} - f\|_1 \le c \cdot \text{OPT}_{\mathcal{C}}(f) + \epsilon_n$$

where as $n \nearrow \infty$, the *statistical rate*, $\epsilon_n \searrow 0$ at a rate independent of $f$, namely, the estimator's error is essentially at most $c$ times the optimal. Since for $f \in \mathcal{C}$ has $\text{OPT}_{\mathcal{C}}(f) = 0$, we see that $\epsilon_n \ge \mathcal{R}_n(\mathcal{C})$ for any estimator.

The key challenge is to obtain such an estimate for dense approximation classes $\mathcal{C}$. One such class is the set of degree-$d$ polynomials, $\mathcal{P}_d$ and its $t$-piecewise extension, $\mathcal{P}_{t,d}$. It is known that by tuning the parameters $t, d$, the bias and variance under $\mathcal{P}_{t,d}$ can be suitably tailored to achieve several in-class minimax rates. For example, if $f$ is a log-concave distribution, choosing $t = n^{1/5}$ and $d = 1$, $\text{OPT}_{\mathcal{P}_{t,d}}(f) + \mathcal{R}_n(\mathcal{P}_{t,d}) = \mathcal{O}(1/n^{2/5})$ [3], matching the minimax rate of learning log-concave distributions. Similarly, minimax rates may be attained for many other structured classes including uni-modal, Gaussian, and mixtures of all three.

The VC dimension, $\text{VC}(\mathcal{C})$, measures the complexity of a class $\mathcal{C}$. For many dense classes, including $\mathcal{P}_{t,d}$, $\mathcal{R}_n(\mathcal{C}) = \Theta(\sqrt{\text{VC}(\mathcal{C})/n})$. For such classes, a cross-validation based estimator $\hat{f}$, such as the minimum distance based selection [6], across a sufficiently fine cover of $\mathcal{C}$, achieves a factor-3 approximation to $\mathcal{C}$,

$$\mathbb{E}\|\hat{f} - f\|_1 \le 3\text{OPT}_{\mathcal{C}}(f) + \mathcal{O}(\sqrt{\text{VC}(\mathcal{C})/n}).$$

However, in general, such methods might have time complexity exponential in $n$. This is especially significant in modern applications that process a large number of samples. [1] provided a near-linear $\mathcal{O}(n \log^3 n)$ time algorithm, ADLS, that still achieves the same factor-3 approximation for $\mathcal{P}_{t,d}$ and the statistical rate $\epsilon_n = \mathcal{O}(\sqrt{t(d+1)/n})$. However it leaves some important questions unanswered.

- **Q1:** ADLS shares the same factor-3 approximation as the generic minimum distance selection. However, for the constant-polynomial class $\mathcal{P}_0$, it is easy to see that the empirical histogram $\hat{f}$ achieves a factor-2 approximation, matching a known lower bound [6]. This raises the question if the factor-3 upper bound can be reduced for higher-degree polynomials as well, and if it can be achieved with statistical rate near the optimal $\sqrt{d(t+1)/n}$.

- **Q2:** ADLS requires prior knowledge of the number $t$ of polynomial pieces, which may be impractical in real applications. Even for structured distribution families, the $t$ achieving their minimax rate can vary significantly. For example, for log-concave distributions, $t = \Theta(n^{1/5})$, and for unimodal distributions, $t = \Theta(n^{1/3})$. This raises the question of whether there are estimators that are optimal for $\mathcal{P}_{t,d}$ simultaneously over all $\forall t \ge 0$.

A partial answer for Q1 was provided in [2] who recently showed that any *finite* class $\mathcal{C}$ can be approximated with the optimal approximation factor of 2, and with statistical rate $\epsilon_n = \tilde{\mathcal{O}}\left(|\mathcal{C}|^{1/5}/n^{2/5}\right)$. While this result can be adapted to infinite classes like $\mathcal{P}_{t,d}$ by constructing finite covers, as Lemma 15 in Appendix E shows, even for the basic single piece quadratic polynomial class $\mathcal{P}_2$, this yields $\epsilon_n = \tilde{\mathcal{O}}(n^{-1/4}) \gg \Theta(n^{-1/2}) = \mathcal{R}_n(\mathcal{P}_2)$. And as with the minimum distance selection discussed above, the result is only information-theoretic without a matching algorithm.

Q2 can be partially addressed by using cross-validation techniques, for example based on the minimum distance selection that compare results for different $t$'s and finds the best. However, as shown in [6], this would add an extra approximation factor of at least 3, and perhaps even 5 as ADLS's estimates are un-normalized, resulting in $c = 5 \cdot 3 = 15$. Furthermore this step raises the statistical rate by an additive $\mathcal{O}(\log n / \sqrt{n})$.

SURF answers both questions in the affirmative. Theorem 1 achieves factors $\le 3$ for all degrees $\le 8$ with optimal $\epsilon_n = \mathcal{O}(\mathcal{R}_n(\mathcal{P}_d))$. Corollary 3 achieves the same factors and a near-optimal $\epsilon_n = \tilde{\mathcal{O}}(\mathcal{R}_n(\mathcal{P}_{t,d}))$ for any $t \ge 0$, even unknown, and runs in time $\mathcal{O}(n \log^2 n)$.

The rest of the paper is organized as follows. In Section 2 we describe the construction of intervals and partitions based on statistically equivalent blocks. In Section 3 we present INT, a polynomial approximation method for any queried interval based on a novel empirical mass interpolation. In Section 4 we explain the MERGE and COMP routines, that respectively combine and compare between piecewise polynomial approximations. We conclude in Section 5 with a detailed comparison of SURF and ADLS, and show experimental results that confirm the theory and show that SURF performs well for a variety of distributions. Proofs of all theorems and lemmas may be found in the supplementary material.

## 1.2 Relation to Prior Work

In terms of objectives, SURF is most closely related to ADLS. Briefly, SURF is simpler, because of which it has a $\mathcal{O}(n \log^2 n)$ time complexity compared to $\mathcal{O}(n \log^3 n)$, it is parallelizable to run in sub-linear time given sorted samples unlike ADLS that uses VC dimension based approaches. As mentioned above, it is also more adaptive. On the other hand, when $t$ is known in advance, ADLS achieves a factor-3 approximation with optimal $\epsilon_n$. For a more detailed comparison, see Section 5.

Among the many other methods that have been employed in distribution estimation, see [16, 5], SURF is inspired by the concept of statistically equivalent blocks introduced in [19, 20]. Distribution estimation methods using this concept partition the domain into regions identified by a fixed number of samples, and perform local estimation on these regions. These methods have the advantage that they are simple to describe, almost always of polynomial time complexity in $n$, and easy to interpret.

The first estimator that used this technique is found in [13]. Expanding on several subsequent works, the notable work [12] shows consistency of a family of equivalent block based estimators for multivariate distributions. See [5] for a more extensive treatment of this subject. Ours is the first work that provides agnostic error guarantees for an equivalent block based estimator.

Other popular estimation methods are the Kernel, nearest neighbor, MLE, and wavelets, see [17]. Another related method uses splines, for example [21, 9]. While MLE and splines may be used for polynomial estimation, MLE is intractable in general, and neither provide agnostic error guarantees.

## 1.3 Main Results

SURF first uses an interpolation routine INT that outputs an estimate, $\hat{f}_{I,\text{INT}} \in \mathcal{P}_d$ for any queried interval $I$. Notice that a degree-$d$ polynomial is determined by the measure it assigns to any $d+1$ distinct sub-intervals of $I$. While ADLS considers fitting the polynomial that minimizes difference in measure to the empirical mass on the worst set of $d+1$ sub-intervals, we show that for low-degree polynomials, it suffices to consider certain special sub-intervals. Provided in Lemma 8, they are functions of $d$ and are sample independent. For $d \leq 8$, the resulting estimate is a factor $< 3$ approximation to $\mathcal{P}_d$, with $\epsilon_n = \mathcal{O}(\mathcal{R}_n(\mathcal{P}_d))$, the optimal statistical rate for $\mathcal{P}_d$.

**Theorem 1.** *Given samples $X^{n-1} \sim f$ for some $n \geq 128$, degree $d$, and an interval $I$ with $n_I$ samples within $I$, INT takes $\mathcal{O}(d^\tau + n_I)$ time, and outputs $\hat{f}_{I,\text{INT}} \in \mathcal{P}_d$ such that*

$$\mathbb{E}\|\hat{f}_{I,\text{INT}} - f\|_I \leq (r_d + 1) \cdot \inf_{h \in \mathcal{P}_d} \|h - f\|_I + r_d \cdot \sqrt{\frac{2(d+1)q_I}{\pi n}},$$

*where $q_I \stackrel{\text{def}}{=} (n_I + 1)/n$, $\|.\|_I$ is the $\ell_1$ norm evaluated on $I$, $\tau < 2.4$ is the matrix inversion exponent, $r_d$ is a fundamental constant whose values are $r_0 = 1, r_1 = 1.25, r_2 \approx 1.42, r_3 \approx 1.55, r_4 \leq 1.675, r_5 \leq 1.774, r_6 \leq 1.857, r_7 \leq 1.930, r_8 \leq 1.999$ for $4 \leq d \leq 8$.*

A few remarks are in order. The additive $\mathcal{O}(\sqrt{q_I/n})$ here is related to the standard deviation in the measure associated with an interval that has $q_I$ fraction of samples. For $d > 8$, $r_d > 3$ and they may be evaluated using Lemma 8.

The main routine of SURF, MERGE, then calls INT to obtain a piecewise estimate for any partition of the domain. MERGE uses COMP to compare between the different piecewise estimates. By imposing a special binary structure on the space of partitions, we allow for COMP to efficiently make this comparison via a divide-and-conquer approach. This allows MERGE, and in turn SURF, to output $\hat{f}_{\text{SURF}}$ in $\mathcal{O}((d^\tau + \log n)n \log n)$ time, where $\tau$ is the matrix inversion exponent. $\hat{f}_{\text{SURF}}$ is a factor-$(r_d + 1)$ approximation for $\mathcal{P}_{t,d}$ $\forall t \geq 0$. The simplicity of SURF, both the polynomial interpolation and divide-and-conquer, allow us to derive all constants explicitly unlike in the previous works. This result is summarized below in Theorem 2 and Corollary 3.

**Theorem 2.** *Given $X^{n-1} \sim f$ for some $n \geq 128$ such that $n$ is a power of 2, and parameters $d \leq 8$, $\alpha > 2$, SURF takes $\mathcal{O}((d^\tau + \log n)n \log n)$ time, and outputs $\hat{f}_{\text{SURF}}$ such that w.p. $\geq 1 - \delta$,*

$$\|\hat{f}_{\text{SURF}} - f\|_1 \leq \min_{\bar{I} \in \Delta_{\mathbb{R}}(X^{n-1})} \sum_{I \in \bar{I}} \left( \frac{(r_d + 1)\alpha}{\alpha - 2} \inf_{h \in \mathcal{P}_d} \|h - f\|_I \right.$$

$$\left. + \frac{r_d(\alpha\sqrt{2} + \sqrt{2} - 1)}{(\sqrt{2} - 1)^2} \sqrt{\frac{5(d+1)q_I \log \frac{n}{\delta}}{n}} \right),$$

*where $q_I$ is the fraction of samples in interval $I$, $\Delta_{\mathbb{R}}(X^n)$ is the collection of all partitions of $\mathbb{R}$ whose intervals start and end at a sample point, $\|\cdot\|_I$ is the $\ell_1$ distance evaluated in interval $I$, $\tau < 2.4$ is the matrix inversion exponent, and $r_d > 0$ is the constant in Theorem 1.*

**Corollary 3.** *Running* SURF *with $d \leq 8$, $\alpha > 4$,*

$$\mathbb{E}\|\hat{f}_{\text{SURF}} - f\|_1 \leq \min_{t \geq 0} \left( (r_d + 1) \left( 1 + \frac{4}{\alpha} \right) \cdot \text{OPT}_{\mathcal{P}_{t,d}}(f) + \tilde{\mathcal{O}} \left( \alpha \sqrt{\frac{t \cdot (d+1)}{n}} \right) \right).$$

## 2 Intervals and Partitions

For $n \geq 1$, let $X^{(n-1)} \stackrel{\text{def}}{=} X_{(1)}, \ldots, X_{(n-1)}$ be the increasingly-sorted values of $X^{n-1}$. For integers $0 \leq a < b \leq n$, these samples define intervals on the real line $\mathbb{R}$,

$$I_{a,b} = (-\infty, X_{(b)}) \text{ if } a = 0, \ I_{a,b} = [X_{(a)}, X_{(b)}) \text{ if } 0 < a < b < n, \ I_{a,b} = [X_{(a)}, \infty) \text{ if } b = n.$$

The *interval-* and *empirical-probabilities* are $P_{a,b} \stackrel{\text{def}}{=} \int_{I_{a,b}} dF$, and $q_{a,b} \stackrel{\text{def}}{=} \frac{b-a}{n}$. For any $0 \leq a < b \leq n$, $I_{a,b}$ forms a *statistically equivalent* block [19], wherein $P_{a,b} \sim \text{Beta}(b-a, n-(b-a))$ regardless of $f$, and $P_{a,b}$ concentrates to $q_{a,b}$.

**Lemma 4.** *For any $0 \leq a < b \leq n$, $\epsilon \geq 0$,*

$$\Pr[|P_{a,b} - q_{a,b}| \geq \epsilon\sqrt{q_{a,b}}] \leq e^{-(n-1)\epsilon^2/2} + e^{-(n-1)\epsilon^2 q_{a,b}/(2q_{a,b}+2\epsilon\sqrt{q_{a,b}})}.$$

We extend this concentration from one interval to many. For a fixed $\epsilon > 0$, let $\mathcal{Q}_\epsilon$ be the event that

$$\forall \, 0 \leq a < b \leq n, \ |P_{a,b} - q_{a,b}| \leq \epsilon\sqrt{q_{a,b}}.$$

**Lemma 5.** *For any $n \geq 128$ and $\epsilon \geq 0$,*

$$P[\mathcal{Q}_\epsilon] \geq 1 - n(n+1)/2 \cdot \left( e^{-(n-1)\epsilon^2/2} + e^{-(n-1)\epsilon^2/(2+2\epsilon\sqrt{n})} \right).$$

Notice that $\mathcal{Q}_\epsilon$ refers to a stronger concentration event that involves $\sqrt{q_{a,b}} \, \forall 0 \leq a < b \leq n$ and standard VC dimension based bounds cannot be readily applied to obtain Lemma 5.

A collection of countably many disjoint intervals whose union is $\mathbb{R}$ is said to be a partition of $\mathbb{R}$. A distribution $\bar{q}$, consisting of interval empirical probabilities is called an *empirical distribution*, or that each probability in $\bar{q}$ is a multiple of $1/n$. The set of all empirical distributions is denoted by $\Delta_{\text{emp},n}$. Since each $q \in \bar{q} \in \Delta_{\text{emp},n} \geq 1/n$, $\bar{q}$ may be split into its finitely many probabilities as $\bar{q} = (q_1, \ldots, q_k)$. These probabilities define a partition if we consider the first increasingly sorted $q_1 n$ samples, the next $q_2 n$ samples and so on. For $1 \leq i \leq k$, let $r_i \stackrel{\text{def}}{=} \sum_{j=1}^{i-1} q_j$ (note that $r_1 = 0$). The empirical distribution defines the following *interval partition*:

$$\bar{I}_{\bar{q}} \stackrel{\text{def}}{=} (I_{r_1 n, (r_1+q_1)n}, I_{r_2 n, (r_2+q_2)n}, \ldots, I_{r_k n, (r_k+q_k)n}).$$

## 3 The Interpolation Routine

This section describes INT, which outputs an estimate $\hat{f}_{I,\text{INT}} \in \mathcal{P}_d$ for any queried interval $I$. WLOG let $I = [0,1]$. A collection, $\bar{n}_d = (n_0, \ldots, n_{d+1})$ such that $0 = n_0 \leq n_1 \leq \cdots \leq n_d \leq n_{d+1} = 1$ is said to be a node partition of $[0,1]$. Let $\mathcal{N}_d$ be the set of node partitions and for the set of non-zero polynomials, $\mathcal{P}_d \setminus \{0\}$, define $r : \mathcal{N}_d, \mathcal{P}_d \to [1, \infty)$ and its suprema

$$r(\bar{n}_d, h) \stackrel{\text{def}}{=} \frac{\int_0^1 |h|}{\sum_{i=1}^{d+1} |\int_{n_{i-1}}^{n_i} h|}, \quad r_d(\bar{n}_d) = \sup_{h \in \mathcal{P}_d \setminus \{0\}} r(\bar{n}_d, h). \tag{1}$$

Notice that $r(\bar{n}_d, h) \geq 1$ since the absolute integral $\geq$ the sum of absolute areas. For any node partition $\bar{n}_d \in \mathcal{N}_d$, let $J_{\bar{n}_d,i} \stackrel{\text{def}}{=} [n_{i-1}, n_i]$, $i \in \{1, \cdots, d+1\}$ so that $\bar{J}_{\bar{n}_d} = (J_{\bar{n}_d,1}, \ldots, J_{\bar{n}_d,d+1})$ partitions $[0,1]$. Let $\hat{f}_{\bar{n}_d} \in \mathcal{P}_d$ be the unique polynomial whose measure on all $d+1$ intervals in $\bar{J}_{\bar{n}_d}$ matches its empirical mass. It is defined as:

$$\hat{f}_{\bar{n}_d} \stackrel{\text{def}}{=} h \in \mathcal{P}_d : \forall i \in \{1, \cdots, d+1\}, \int_{n_{i-1}}^{n_i} h(z)dz = q_{J_{\bar{n}_d,i}}, \tag{2}$$

where for $n_J$ samples that lie within an interval $J$, $q_J \stackrel{\text{def}}{=} (n_J + 1)/n$. Computation of $\hat{f}_{\bar{n}_d}$ involves a calculation of $d+1$ empirical masses that takes $\mathcal{O}(n_J)$ time, and solving a system of $d+1$ linear equations that takes $\mathcal{O}(d^\tau)$ time, where $\tau < 2.4$ is the matrix inversion exponent, for a $\mathcal{O}(n_J + d^\tau)$ run time. The estimate $\hat{f}_{\bar{n}_d}$ corresponding to any choice of $\bar{n}_d \in \mathcal{N}_d$ satisfies the following:

**Lemma 6.** *For interval $I = [0,1]$ with empirical probability $q_I$, any $\bar{n}_d \in \mathcal{N}_d$, and $\epsilon > 0$, the estimate $\hat{f}_{\bar{n}_d}$ (2) is such that under event $\mathcal{Q}_\epsilon$,*

$$\|\hat{f}_{\bar{n}_d} - f\|_1 \le (1 + r_d(\bar{n}_d)) \inf_{h \in \mathcal{P}_d} \|h - f\|_1 + r_d(\bar{n}_d)\epsilon\sqrt{(d+1)q_I}.$$

In Lemma 7, we show that for any $\bar{n}_d \in \mathcal{N}_d$, there exists an $r_d(\bar{n}_d)$ achieving $h \in \mathcal{P}_d$, and that it belongs to a special set, $\mathcal{P}_{\bar{n}_d} \subseteq \mathcal{P}_d$,

$$\mathcal{P}_{\bar{n}_d} \overset{\text{def}}{=} \Big\{ h \in \mathcal{P}_d : \exists i_1 \in \{1, \dots, d+1\} : \forall i \in \{1, \dots, d+1\} \setminus \{i_1\}, \int_{n_{i-1}}^{n_i} h = 0 \Big\}.$$

In words, $\mathcal{P}_{\bar{n}_d}$ is the set of polynomials that has a non-zero area in at most one $I \in \bar{I}_{\bar{n}_d}$.

**Lemma 7.** *For any degree-$d$ and $\bar{n}_d \in \mathcal{N}_d$,*

$$r_d(\bar{n}_d) = \sup_{h \in \mathcal{P}_d} r(\bar{n}_d, h) = \max_{h \in \mathcal{P}_{\bar{n}_d}} r(\bar{n}_d, h).$$

Let the smallest $r_d(\bar{n}_d)$ be denoted by $r_d^\star \overset{\text{def}}{=} \inf_{\bar{n}_d \in \mathcal{N}_d} r_d(\bar{n}_d)$. Lemma 8 shows that there exists an $\bar{n}_d$ that attains the infimum. It is denoted by $\bar{n}_d^\star = \arg\min_{\bar{n}_d \in \mathcal{N}_d} r_d(\bar{n}_d)$. For $d \le 3$, we calculate $r_d^\star$ and $\bar{n}_d^\star$. For $4 \le d \le 8$ we find a $\bar{n}_d \in \mathcal{N}_d$ such that the corresponding $r_d(\bar{n}_d) < 2$.

**Lemma 8.** *For $d \le 3$, there exists a node collection $\bar{n}_d^\star$ that achieves $r_d^\star$. These, and their respective $r_d^\star$ are given by*

| $d$ | $\bar{n}_d^\star$ | $r_d^\star$ |
|---|---|---|
| 0 | $(0,1)$ | 1 |
| 1 | $(0,0.5,1)$ | 1.25 |
| 2 | $\approx (0, 0.2599, 0.7401, 1)$ | $\approx 1.42$ |
| 3 | $\approx (0, 0.1548, 0.5, 0.8452, 1)$ | $\approx 1.56$ |

Denoting $\bar{n}_2^\star = (0, \alpha_0, 1 - \alpha_0, 1)$, and $\bar{n}_3^\star = (0, \beta_0, 0.5, 1 - \beta_0, 1)$, the exact values of $\alpha_0$, $\beta_0$, are obtained as roots to a degree-$14$ and degree-$69$ polynomial that we explicitly provide. For degrees $4 \le d \le 8$, the following $\bar{n}_d \in \mathcal{N}_d$ and $r_d(\bar{n}_d)$ provide upper bounds on $r_d^\star$.

| $d$ | $\bar{n}_d$ | $r_d(\bar{n}_d)$ |
|---|---|---|
| 4 | $(0, 0.1015, 0.348, 0.652, 0.8985, 1)$ | $< 1.675$ |
| 5 | $(0, 0.071, 0.254, 0.5, 0.746, 0.929, 1)$ | $< 1.774$ |
| 6 | $(0, 0.053, 0.192, 0.390, 0.610, 0.808, 0.947, 1)$ | $< 1.857$ |
| 7 | $(0, 0.0405, 0.149, 0.310, 0.5, 0.690, 0.851, 0.9595, 1)$ | $< 1.930$ |
| 8 | $(0, 0.032, 0.119, 0.252, 0.414, 0.586, 0.749, 0.881, 0.968, 1)$ | $< 1.999$ |

For a given interval $I$ and $d \le 8$, INT first scales and shifts $I$ to obtain $[0,1]$. It then constructs $\hat{f}_{\bar{n}_d}$ using the $\bar{n}_d$ in Lemma 8. The output $\hat{f}_{I,\text{INT}}$ is the re-scaled-shifted $\hat{f}_{\bar{n}_d}$.

## 4 The Compare and Merge Routines

This section presents MERGE and COMP, the main routines of SURF. For any contiguous collection of intervals $\bar{I}$, let $\hat{f}_{\bar{I},\text{INT}}$ be the piecewise polynomial estimate consisting of $\hat{f}_{I,\text{INT}} \in \mathcal{P}_d$ given by INT in each $I \in \bar{I}$. The key idea in SURF is to separate interval partitions into a binary hierarchy, effectively allowing a comparison of all the superpolynomially many (in $n$) estimates corresponding to the different interval partitions, but by using only $\tilde{\mathcal{O}}(n)$ comparisons.

Recall that $n$ here a power of 2 and define the integer $D \overset{\text{def}}{=} \log_2 n$. An empirical distribution, $\bar{q} \in \Delta_{\text{emp,n}}$, is called a *binary* distribution if each of its probability values take the form $1/2^d$, for some integer $0 \le d \le D$. The corresponding interval partition, $\bar{I}_{\bar{q}}$, is said to be a *binary partition*.

$$\Delta_{\text{bin,n}} \overset{\text{def}}{=} \{\bar{q} \in \Delta_{\text{emp,n}} : \forall q \in \bar{q}, q = 1/2^{\nu(q)}, 0 \le \nu(q) \le D, \nu(q) \in \mathbb{Z}\}.$$

For example $\bar{q} = (1)$, $\bar{q} = (1/2, 1/4, 1/4)$, $\bar{q} = (1/4, 1/8, 1/8, 1/2)$ are binary distributions. Similarly, $(1/n, \dots, 1/n) = (1/2^{\log n}, \dots, 1/2^{\log n})$ is also a binary distribution since $n$ here is a power of 2 (assume $n \ge 8$ so that they are all in $\Delta_{\text{emp,n}}$). Lemma 9 shows that $\Delta_{\text{bin,n}}$ retains most of the approximating power of $\Delta_{\text{emp,n}}$. In particular, that for any $\bar{q} \in \Delta_{\text{emp,n}}$, there exists a binary distribution $\bar{q}' \in \Delta_{\text{bin,n}}$ such that $\bar{I}_{\bar{q}'}$ has a smaller bias than $\bar{I}_{\bar{q}}$, while its deviation under the concentration event, $\mathcal{Q}_\epsilon$, is larger by less than a factor of $1/(\sqrt{2} - 1)$.

**Lemma 9.** *For any empirical distribution $\bar{q} \in \Delta_{\mathrm{emp,n}}$, there exists $\bar{q}' \in \Delta_{\mathrm{bin,n}}$ such that*

$$\|f^\star_{\bar{I}_{\bar{q}'}} - f\|_1 \le \|f^\star_{\bar{I}_{\bar{q}}} - f\|_1, \quad \sum_{q \in \bar{q}'} \epsilon\sqrt{q} \le \sum_{q \in \bar{q}} \frac{1}{\sqrt{2}-1} \epsilon\sqrt{q},$$

*where for any $d > 0$, $f^\star_{\bar{I}}$ is the piecewise degree-$d$ polynomial closest to $f$ on the partition $\bar{I}$.*

For a fixed $\bar{p} \in \Delta_{\mathrm{bin,n}}$, let $\Delta_{\mathrm{bin,n},\le\bar{p}}$ be the set of binary distributions such that for any $\bar{q} \in \Delta_{\mathrm{bin,n},\le\bar{p}}$, each $I_1 \in \bar{I}_{\bar{q}}$ is contained in some $I_2 \in \bar{I}_{\bar{p}}$.

$$\Delta_{\mathrm{bin,n},\le\bar{p}} \overset{\mathrm{def}}{=} \{\bar{q} \in \Delta_{\mathrm{bin,n}} : \forall I_1 \in \bar{I}_{\bar{q}}, \exists I_2 \in \bar{I}_{\bar{p}}, I_1 \subseteq I_2\}. \tag{3}$$

For example if $\bar{p} = (1/2, 1/4, 1/4)$ is the binary distribution, $(1/4, 1/4, 1/8, 1/8, 1/4)$, $(1/2, 1/4, 1/8, 1/8) \in \Delta_{\mathrm{bin,n},\le\bar{p}}$, whereas $(1/2, 1/2) \notin \Delta_{\mathrm{bin,n},\le\bar{p}}$.

### 4.1 The MERGE Routine

The MERGE routine operates in $i \in \{1, \ldots, D\}$ steps (recall $D = \log_2 n$) where at the end of each step $i$, MERGE holds onto a binary distribution $q_i$. At at the last step $i = D$, SURF outputs the piecewise estimate on the partition given by $\bar{q}_D$, i.e. $\hat{f}_{\mathrm{SURF}} = \hat{f}_{\bar{I}_{\bar{q}_D},\mathrm{INT}}$. Let

$$D(i) \overset{\mathrm{def}}{=} D - i \text{ and let } \bar{u}_i \overset{\mathrm{def}}{=} \left(1/2^{D(i)}, \ldots, 1/2^{D(i)}\right).$$

Initialize $\bar{q}_0 \leftarrow (1/n, \ldots, 1/n)$. Start with $i = 1$ and assign $\bar{s} \leftarrow \bar{q}_{i-1}$. Throughout its run MERGE maintains $\bar{s} = \bar{q}_{i-1} \in \Delta_{\mathrm{bin,n},\le\bar{u}_i}$. For instance this holds for $i = 1$ since $\bar{u}_1 = (2/n, \ldots, 2/n)$. MERGE considers merging the probability values in $\bar{s}$ to match it with $\bar{u}_i$. For example if at step $i = D - 1$, $\bar{s} = (1/8, 1/8, 1/4, 1/4, 1/4)$, it considers merging $(1/8, 1/8, 1/4)$ and $(1/4, 1/4)$ to obtain $\bar{u}_{D-1} = (1/2, 1/2)$.

This decision is made by invoking the COMP routine on intervals corresponding to the merged probability value. In this case COMP is called on intervals $\bar{I} \in \bar{I}_{\bar{s}}$ corresponding to $(1/8, 1/8, 1/4)$ and $(1/4, 1/4)$ respectively, along with the tuning parameter $\gamma$,

$$\gamma \overset{\mathrm{def}}{=} \alpha \cdot r_d \cdot \epsilon\sqrt{d+1}.$$

While COMP decides to merge depending on the increment in bias on the merged interval versus the decrease in variance, $\gamma$ tunes this trade-off. A large $\gamma$ results in a decision to merge while a small $\gamma$ has the opposite effect. If $\mathrm{COMP}(\bar{I}, \gamma) \le 0$ the probabilities in $\bar{s}$ corresponding to $\bar{I}$ are merged and copied into $\bar{q}_i$. Otherwise they are copied as is into $\bar{q}_i$. See Appendix F.1 for a detailed description.

At each step $i \in \{1, \ldots, D\}$, MERGE calls COMP on $2^{D(i)}$ intervals, each consisting of $2^i$ samples. Thus each step of MERGE takes $\mathcal{O}(2^{D(i)} \cdot (d^\tau + \log(2^i)) \cdot 2^i) = \mathcal{O}((d^\tau + \log n)2^D)$ time. The total time complexity is therefore $\mathcal{O}((d^\tau + \log n)2^D D) = \mathcal{O}((d^\tau + \log n)n \log n)$.

### 4.2 The COMP Routine

COMP receives an interval partition $\bar{I}$ consisting of $m$ samples and the parameter $\gamma$ as input, and returns a real value that indicates its decision to merge the probabilities under $\bar{I}$. Let $\bar{s}$ be the set of empirical probabilities corresponding to $\bar{I}$. Let the merged interval be $I$ and let $\hat{f} = \hat{f}_{\mathrm{INT},I}$ be the polynomial estimate on $I$.

For simplicity suppose $\bar{I} = (I_1, I_2)$ with empirical mass $s_{I_1}, s_{I_2}$ respectively, and let $\mathrm{OPT}_{I,\mathcal{P}_{t,d}}(f) = \min_{h \in \mathcal{P}_{t,d}} \|h - f\|_I$. If $\bar{I}$ is merged, observe that the bias $\mathrm{OPT}_{I,\mathcal{P}_{t,d}}(f) \ge \mathrm{OPT}_{I_1,\mathcal{P}_{t,d}}(f) + \mathrm{OPT}_{I_2,\mathcal{P}_{t,d}}(f)$ increases but since $s_I = s_{I_1} + s_{I_2}$, $\sqrt{s_I} \le \sqrt{s_{I_1}} + \sqrt{s_{I_2}}$, resulting in a smaller $\epsilon$-deviation under event $\mathcal{Q}_\epsilon$ in Lemma 5. Consider their difference parameterized by the constant $\gamma$,

$$\mu'_\gamma(f) \overset{\mathrm{def}}{=} (\mathrm{OPT}_{I_1,\mathcal{P}_{t,d}}(f) + \mathrm{OPT}_{I_2,\mathcal{P}_{t,d}}(f) - \mathrm{OPT}_{I,\mathcal{P}_{t,d}}(f)) - \gamma(\sqrt{s_{I_1}} + \sqrt{s_{I_2}} - \sqrt{s_I}).$$

If $\mu'_\gamma(f) \le 0$, it indicates that the overall $\ell_1$ error is smaller under the merged $I$. While $\mu'_\gamma(f)$ cannot be evaluated without access to the underlying $f$, we use a proxy, $\mu_{\bar{I},\gamma}(f)$ that is defined next.

Normalize $\bar{s}$ so that it is a distribution, and consider $\bar{p} \in \Delta_{\mathrm{bin,m}}$ such that $\bar{s} \in \Delta_{\mathrm{bin,m},\le\bar{p}}$ and the piecewise estimate on $\bar{I}_{\bar{p}}$, i.e. $\hat{f}_{\bar{I}_{\bar{p}},\mathrm{INT}}$. Define $\Lambda_{\bar{I}_{\bar{p}}}(\hat{f}) \overset{\mathrm{def}}{=} \|\hat{f}_{\bar{I}_{\bar{p}},\mathrm{INT}} - \hat{f}\|_{\bar{I}_{\bar{p}}}$, $\lambda_{\bar{p},\gamma} \overset{\mathrm{def}}{=} \sum_{p \in \bar{p}} \gamma\sqrt{p}$,

$$\mu_{\bar{I}_{\bar{s}},\gamma}(\hat{f}) \overset{\mathrm{def}}{=} \max_{\bar{p}:\bar{s}\in\Delta_{\mathrm{bin,m},\le\bar{p}}} \Lambda_{\bar{I}_{\bar{p}}}(\hat{f}) - \lambda_{\bar{p},\gamma}.$$

COMP returns $\mu_{\bar{I}_{\bar{s}},\gamma}(\hat{f})$ via a divide-and-conquer based implementation, and results in $\mathcal{O}((d^\tau + \log m)m)$ time. A detailed description is provided in Appendix F.3. Lemma 10 shows that under event $\mathcal{Q}_\epsilon$, $\hat{f}_{\mathrm{SURF}}$ is within a constant factor of the best piecewise polynomial approximation over any binary partition, plus its deviation in probability under $\mathcal{Q}_\epsilon$ times $\mathcal{O}(\sqrt{d+1})$.

**Lemma 10.** *Given samples $X^{n-1} \sim f$, for some $n$ that is a power of 2, degree $d \le 8$ and the threshold $\alpha > 2$, SURF outputs $\hat{f}_{SURF}$ in time $\mathcal{O}((d^\tau + \log n)n \log n)$ such that under event $\mathcal{Q}_\epsilon$,*

$$\|\hat{f}_{SURF} - f\|_1 \le \min_{\bar{p} \in \Delta_{\mathrm{bin,n}}(X^{n-1})} \sum_{I \in \bar{I}_{\bar{p}}} \left( \frac{(r_d+1)\alpha}{\alpha-2} \inf_{h \in \mathcal{P}_d} \|h-f\|_I \right.$$
$$\left. + \frac{r_d(\alpha\sqrt{2} + \sqrt{2}-1)}{\sqrt{2}-1} \epsilon \sqrt{(d+1)q_I} \right),$$

*where $q_I$ is the empirical mass under interval $I$, $r_d$ is the constant in Theorem 1.*

## 5 Comparison and Experiments

We compare the factor improvement of SURF with ADLS, expand on larger degrees-$d$ polynomial approximation, and in particular, address learning Gaussians optimally. We also describe how SURF benefits from its local nature, enabling a distributed computation. Our experiments show that SURF is more adaptive than ADLS, and perform additional experiments on both synthetic and real datasets.

The following table compares SURF with ADLS in terms of the expected error. For $d \le 8$, $r_d \in [2,3)$ is the factor in Theorem 1, and $\tau, \omega \in [2, 2.4]$ are constants. We achieve a lesser factor approximation at nearly the optimal statistical rate, with an improved time complexity in both $n$ and $d$.

| | SURF | ADLS |
|---|---|---|
| $\mathcal{P}_d$ | $r_d \mathrm{OPT}_{\mathcal{P}_{t,d}}(f) + \sqrt{\frac{2d}{\pi n}}$ | $3\mathrm{OPT}_{\mathcal{P}_{t,d}}(f) + \mathcal{O}\left(\sqrt{\frac{d}{n}}\right)$ |
| $\mathcal{P}_{t,d}$ and known $t$ | $r_d \mathrm{OPT}_{\mathcal{P}_{t,d}}(f) + \mathcal{O}\left(\sqrt{\frac{t(d+1)\log n}{n}}\right)$ | $3\mathrm{OPT}_{\mathcal{P}_{t,d}}(f) + \mathcal{O}\left(\sqrt{\frac{t(d+1)}{n}}\right)$ |
| $\mathcal{P}_{t,d}$ and unknown $t$ | $r_d \min_{t \ge 0} \left( \mathrm{OPT}_{\mathcal{P}_{t,d}}(f) + \mathcal{O}\left(\sqrt{\frac{t(d+1)\log n}{n}}\right) \right)$ | $15 \min_{t \ge 0} \left( \mathrm{OPT}_{\mathcal{P}_{t,d}}(f) + \mathcal{O}\left(\sqrt{\frac{t(d+1)}{n}}\right) \right) + \mathcal{O}\left(\frac{\log n}{\sqrt{n}}\right)$ |
| Time complexity | $\mathcal{O}(n \log^2 n d^\tau)$ | $\mathcal{O}(n \log^3 n d^{3+\omega})$ |

While for $d > 8$, SURF does not improve the approximation factor below $< 3$, we note that polynomial approximations of larger degrees exhibit oscillatory behavior, for example around the edges when approximating a pulse. Called the Runge phenomenon [18], this may result in an unbounded $\ell_p$ distance for $p > 1$. In this scenario it may be preferred to use a lower degree polynomial, but with an appropriately large $t$. Consider the important case when $f$ is a Gaussian distribution. As shown in Lemma 16, $\mathrm{OPT}_{\mathcal{P}_{t,d}}(f) = \mathcal{O}(1/t^{d-1})$. Using the fact that $\epsilon_n = \tilde{\mathcal{O}}(\sqrt{t(d+1)/n})$ and minimizing $\mathrm{OPT}_{\mathcal{P}_{t,d}}(f) + \epsilon_n$ over $t$ for a fixed $d$, we obtain $\|\hat{f}_{\mathrm{SURF}} - f\|_1 = \tilde{\mathcal{O}}((d+1)/n)^{\frac{1}{2} - \frac{1}{4d-2}}$. Even for an astronomical $n = 2^{100}$ samples, choosing $d = 8$ ensures that $n^{\frac{1}{4d-2}} \le 11$. Thus in almost all scenarios of practical interest we nearly match (upto a $\sqrt{\log n}$ factor) the minimax rate $\mathcal{O}(1/n)^{\frac{1}{2}}$ of learning Gaussians. While ADLS avoids this factor of $n^{\frac{1}{4d-2}}$, they do so by using $d = \mathcal{O}(\log n)$ which may present the above drawbacks. For degrees that are even larger, the $\Omega(d^5 n \log^3 n)$ time taken by ADLS may make it impractical.

In terms of time complexity, SURF benefits from its local nature, enabling a distributed computation. As detailed in Appendix F, if provided with pre-sorted samples, a known $t$ and memory $m \ge t$, it can be adapted to run in time $\mathcal{O}((d^\tau + \log n)n \max\{1/t, \log n/m\}) \ll \mathcal{O}(n)$, if $t \approx n$. We now follow up with an experimental comparison. SURF is run with $\alpha = 0.25$ and the errors are averaged over 10 runs. In running ADLS we use the provided code as is. Figure 1 compares the $\ell_1$ error in piecewise-linear estimation using SURF vs ADLS on the distributions considered in [1], namely, a beta, Gamma, and Gaussian mixture. The plots correspond to the errors incurred on running SURF, and ADLS with pieces $t = 5, 10, 20, 40, 60$. While some hyperparameter optimizations may aid

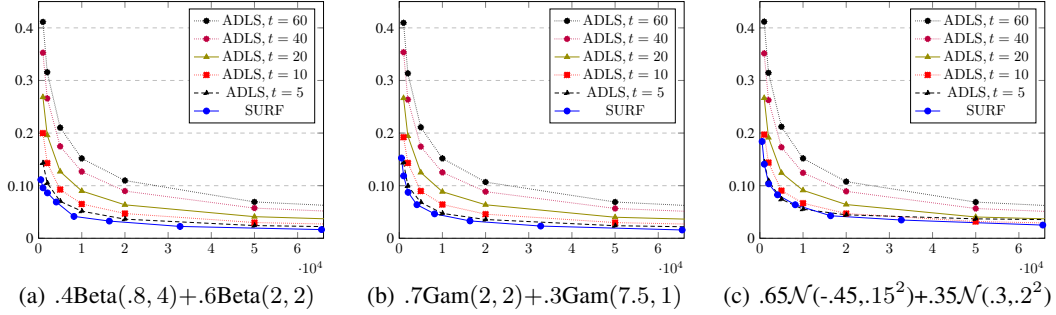

(a) .4Beta(.8, 4)+.6Beta(2, 2)  (b) .7Gam(2, 2)+.3Gam(7.5, 1)  (c) .65$\mathcal{N}$(-.45,.15²)+.35$\mathcal{N}$(.3,.2²)

Figure 1: $\ell_1$ error versus number of samples of piece-wise linear SURF and ADLS.

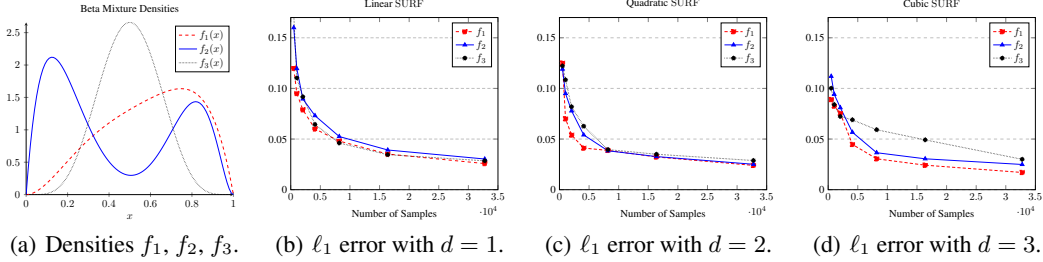

(a) Densities $f_1, f_2, f_3$.  (b) $\ell_1$ error with $d = 1$.  (c) $\ell_1$ error with $d = 2$.  (d) $\ell_1$ error with $d = 3$.

Figure 2: Evaluation of the estimate output by SURF with degrees $d = 1, 2, 3$, $\alpha = 0.25$, on $f_1 = 0.4\text{Beta}(3, 4) + 0.6\text{Beta}(5, 2)$, $f_2 = 0.4\text{Beta}(10, 3) + 0.6\text{Beta}(2, 8)$, and $f_3 = \text{Beta}(6, 6)$.

either algorithms, observe that the errors can be much larger with the wrong $t$. Significantly, the $t = 5$ for which the results are comparable, is also roughly the number of pieces that SURF outputs.

Experiments show that SURF learns a wide range of parametric families such as the beta, Gaussian and Gamma distributions. In Figure 2 we show results on beta mixture distributions over $[0, 1]$, as they accommodate a wide range of shapes. Other results may be found in Appendix G. Let Beta($\alpha, \beta$) be the beta density with parameters $\alpha, \beta$. We run SURF to estimate three distributions, as shown in Figure 2(a). SURF estimates them using piecewise polynomials of degree $d = 1, 2, 3$. Figures 2(b)–2(d) show the resulting $\ell_1$ errors. Observe that the errors are decaying, and are similar between distributions. This is not surprising since low degree polynomial approximations largely rely on local smoothness, which all of the considered densities possess. By the same reasoning, on increasing $d$ from 1 to 3, the variation in error between distributions increases. The smoother $f_1$ starts incurring a smaller $\ell_1$ error than $f_2$ and $f_3$.

Next, we run SURF with $d = 2$ to estimate $f = 0.3f_{\text{Beta},3,10} + 0.7f_{\text{Beta},17,4}$ with $n = 1024, 4096, 16384, 65536$. Figure 3 plots the resulting estimates against $f$. Notice that the estimate not only successively better estimates $f$ in $\ell_1$ distance, but also pointwise converges to $f$.

Finally, we ran SURF on real data sets consisting of salaries from the 1994 US census and electric signals from the sensorless drive diagnosis dataset [8], that have been used to evaluate classification

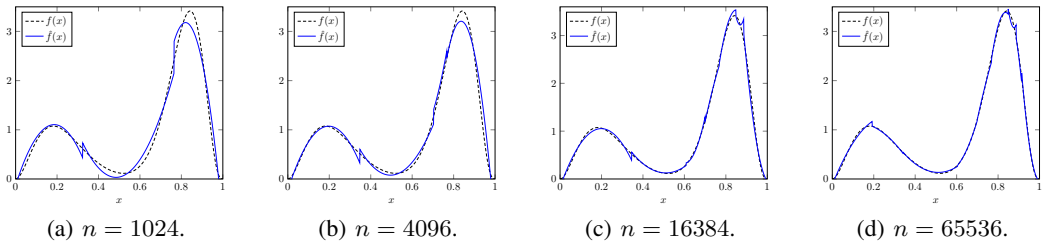

(a) $n = 1024$.  (b) $n = 4096$.  (c) $n = 16384$.  (d) $n = 65536$.

Figure 3: SURF with degree $d = 2$, $\alpha = 0.25$ estimating $f = 0.3f_{\text{Beta},3,10} + 0.7f_{\text{Beta},17,4}$ with $n = 1024, 4096, 16384, 65536$ samples.

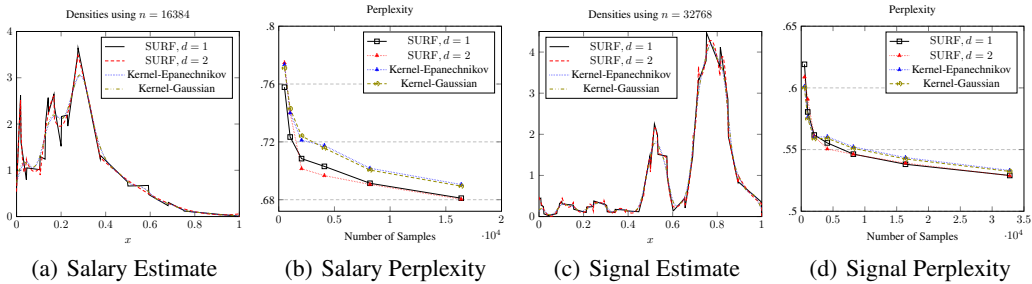

|  (a) Salary Estimate | (b) Salary Perplexity | (c) Signal Estimate | (d) Signal Perplexity |

Figure 4: Real data estimates and perplexity of SURF vs MLE based Kernel estimators

algorithms [10, 4, 14]. We trim $0.5\%$ of samples on either side and re-scale to obtain 57923 samples that lie in $[0, 1]$. Figures 4(a) and 4(c) show the estimate output by SURF and the similarly non-parametric, popularly used Kernel estimator with Epanechnikov and Gaussian kernels via the `fitdist()` function in $\text{MATLAB}^{\circledR}$. As it can be observed, SURF, without any hidden parameter, recovers characteristic features of the distribution such as the clusters, mode values, and tails. This is in contrast with ADLS, that, strictly speaking, cannot be used in this context as it requires additional cross-validation to tune $t$ based on the number of clusters, etc. The perplexity, or the exponent of the average negative log-likelihood on unseen samples, is a commonly used measure in practice to evaluate an estimate. Figures 4(d), 4(b) compares the perplexity on a test set with one-fourth the number of samples. As it can be seen, even as `fitdist()` outputs the perplexity minimizer on the training set, SURF performs better.

## Broader Impact

SURF is a simple, universal, robust, and fast algorithm for the important problem of estimating distributions by piecewise polynomials. Real-life applications are likely to be approximated by relatively low-degree polynomials and require fast algorithms. SURF is particularly well-suited for these regimes.

## Acknowledgments

We are grateful to the National Science Foundation (NSF) for supporting this work through grants CIF-1564355 and CIF-1619448.

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
