[Supplementary Material]

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

# A  Introduction

## A.1  Proof of Theorem 1

**Proof**  For a given $d$, INT outputs $\hat{f}_{I,\text{INT}}$, the re-scaled-shifted $\hat{f}_{\bar{n}_d}$ given by the corresponding $\bar{n}_d \in \mathcal{N}_d$ in Lemma 8. Choosing $\epsilon(\delta) = \sqrt{5\log(1/\delta)/n}$, for $n \geq 128$, $\mathcal{Q}_{\epsilon(\delta)}$ occurs with probability $\geq 1 - \delta$ from Lemma 5. Using Lemma 6 with $\epsilon(\delta)$ completes the proof.

## A.2  Proof of Theorem 2

**Proof**  Choosing $\epsilon(\delta) = \sqrt{5\log(n/\delta)/n}$, for $n \geq 128$, $\mathcal{Q}_{\epsilon(\delta)}$ occurs with probability $\geq 1 - \delta$ from Lemma 5. Using Lemma 9 on top of Lemma 10 proves the theorem.

## A.3  Proof of Corollary 3

**Proof**  From Theorem 2, w.p. $\geq 1 - \delta$,

$$
\begin{aligned}
\|\hat{f}_{\mathcal{A}(X^{n-1})} - f\|_1 &\leq \min_{\bar{I} \in \Delta_{\mathbb{R}}(X^{n-1})} \sum_{I \in \bar{I}} \left( \frac{(r_d+1)\cdot\alpha}{\alpha - 2} \cdot \inf_{h \in \mathcal{P}_d} \|h - f\|_I \right. \\
&\qquad\qquad \left. + \frac{r_d \cdot (\alpha\sqrt{2} + \sqrt{2} - 1)}{(\sqrt{2}-1)^2} \sqrt{\frac{5(d+1)q_I \log\frac{n}{\delta}}{n}} \right) \\
&\overset{(a)}{\leq} \min_{t \geq 0} \left( \frac{(r_d+1)\cdot\alpha}{\alpha-2}\text{OPT}_{\mathcal{P}_{t,d}} + \frac{r_d \cdot (\alpha\sqrt{2}+\sqrt{2}-1)}{(\sqrt{2}-1)^2}\sqrt{\frac{5t\cdot(d+1)\log\frac{n}{\delta}}{n}} \right) \\
&\overset{(b)}{\leq} \min_{t \geq 0} \left( (r_d+1)\left(1 + \frac{4}{\alpha}\right) \cdot \text{OPT}_{\mathcal{P}_{t,d}}(f) \right. \\
&\qquad\qquad \left. + \frac{r_d \cdot (\alpha\sqrt{2}+\sqrt{2}-1)}{(\sqrt{2}-1)^2}\sqrt{\frac{5t\cdot(d+1)\log\frac{n}{\delta}}{n}} \right),
\end{aligned}
$$

where $(a)$ follows since for any partition with $t$ pieces, $\sum_{I\in\bar{I}} \sqrt{q_I} \leq \sqrt{t}$, and $(b)$ follows since for any $x > 4$ $x/(x-2) < 1 + 4/x$. Letting $\alpha \to \infty$ and choosing $\delta \approx 1/n$ completes the proof.

# B  Intervals and Partitions

## B.1  Proof of Lemma 4

**Proof**  For simplicity, let

$$
X \overset{\text{def}}{=} P_{a,b}, \quad p \overset{\text{def}}{=} q_{a,b}
$$

so that $X = P_{a,b} \sim \text{Beta}(nq_{a,b}, n(1 - q_{a,b})) = \text{Beta}(np, n(1-p))$. For any $x, y \in \mathbb{R}^+$, let $\text{B}(x,y) = \Gamma(x)\Gamma(y)/\Gamma(x+y)$ denote the beta function and let $a, b > 0$ and $x \in [0,1]$,

$$
I_x(a,b) \overset{\text{def}}{=} \int_0^x \frac{z^{a-1}(1-z)^{b-1}}{\text{B}(a,b)} dz
$$

be the incomplete beta function. Then,

$$
\begin{aligned}
\Pr[X \leq p - \epsilon\sqrt{p}] &\overset{(a)}{=} I_{p-\epsilon\sqrt{p}}(np, n(1-p)) \\
&\overset{(b)}{=} \sum_{i=np}^{n-1} \binom{n-1}{i}(p - \epsilon\sqrt{p})^i (1 - p + \epsilon\sqrt{p})^{n-1-i} \\
&\overset{(c)}{\leq} e^{-(n-1)D(p\|p-\epsilon\sqrt{p})} \\
&\overset{(d)}{\leq} e^{-(n-1)\frac{\epsilon^2}{2}},
\end{aligned}
$$

where $(a)$ follows by definition, $(b)$ follows by the property of incomplete beta function [7], $(c)$ follows from the Chernoff bound applied to the right tail of a $\mathrm{Binom}(n, p - \epsilon\sqrt{p})$ random variable, and $(d)$ follows since $D(x||y) \leq (x-y)^2/\max\{x,y\}$. Similarly,

$$\Pr[X \geq p + \epsilon\sqrt{p}] \overset{(a)}{=} 1 - I_{(p+\epsilon\sqrt{p})}(np, n(1-p))$$

$$\overset{(b)}{=} 1 - \sum_{i=np}^{n-1} \binom{n-1}{i}(p + \epsilon\sqrt{p})^i(1 - p - \epsilon\sqrt{p})^{n-1-i}$$

$$\leq \sum_{i=0}^{np} \binom{n-1}{i}(p + \epsilon\sqrt{p})^i(1 - p - \epsilon\sqrt{p})^{n-1-i}$$

$$\overset{(c)}{\leq} e^{-(n-1)D(p||p+\epsilon\sqrt{p})}$$

$$\overset{(d)}{\leq} e^{-(n-1)\frac{\epsilon^2 p}{2(p+\epsilon\sqrt{p})}},$$

where $(a)$ follows by definition, $(b)$ follows by the property of incomplete beta function [7], and $(c)$ follows from Chernoff bound applied to the left tail of a $\mathrm{Binom}(n, p + \epsilon\sqrt{p})$ random variable, and $(d)$ follows since $D(x||y) \leq (x-y)^2/\max\{x,y\}$.

## B.2 Proof of Lemma 5

**Proof**    From using the union bound, we have

$$1 - \Pr[\mathcal{Q}_\epsilon] = \Pr\left[\exists 0 \leq a < b \leq n : |P_{a,b} - q_{a,b}| \geq \epsilon\sqrt{q_{a,b}}\right]$$

$$\leq \sum_{0 \leq a < b \leq n} \Pr\left[|P_{a,b} - q_{a,b}| \geq \epsilon\sqrt{q_{a,b}}\right]$$

$$\overset{(a)}{\leq} \sum_{0 \leq a < b \leq n} \left(e^{\frac{-(n-1)\epsilon^2}{2}} + e^{\frac{-(n-1)\epsilon^2 q_{a,b}}{2(q_{a,b}+\epsilon\sqrt{q_{a,b}})}}\right)$$

$$\overset{(b)}{=} \frac{n(n+1)}{2}\left(e^{\frac{-(n-1)\epsilon^2}{2}} + e^{\frac{-(n-1)\epsilon^2 q_{a,b}}{2(q_{a,b}+\epsilon\sqrt{q_{a,b}})}}\right)$$

$$\overset{(c)}{\leq} \frac{n(n+1)}{2}\left(e^{\frac{-(n-1)\epsilon^2}{2}} + e^{-\frac{(n-1)\epsilon^2}{2(1+\epsilon\sqrt{n})}}\right),$$

where $(a)$ follows from Lemma 4, $(b)$ follows since $|0 \leq a < b \leq n| = \binom{n+1}{2}$, and $(c)$ follows since $q_{a,b} \geq 1/n$.

## C    The Interpolation Routine

### C.1    Proof of Lemma 6

**Proof**    For a partition $\bar{I}$ of $I = [0,1]$, and integrable functions $g_1, g_2$, define the distance

$$d_{\bar{I}}(g_1, g_2) \overset{\text{def}}{=} \sum_{J \in \bar{I}} \left|\int_J g_1 - \int_J g_2\right|. \tag{4}$$

In words, $d_{\bar{I}}(g_1, g_2)$ is the sum of absolute differences between measures under $g_1$ and $g_2$ across all intervals in $\bar{I}$. For any $h \in \mathcal{P}_d$,

$$
\begin{aligned}
\|\hat{f}_{\bar{n}_d} - f\|_I &\leq \|h - f\|_I + \|\hat{f}_{\bar{n}_d} - h\|_I \\
&\stackrel{(a)}{\leq} \|h - f\|_I + r_d(\bar{n}_d) d_{\bar{I}_{\bar{n}_d}}(h, \hat{f}_{\bar{n}_d}) \\
&\stackrel{(b)}{\leq} \|h - f\|_I + r_d(\bar{n}_d) \left( d_{\bar{I}_{\bar{n}_d}}(h, f) + d_{\bar{I}_{\bar{n}_d}}(f, \hat{f}_{\bar{n}_d}) \right) \\
&\stackrel{(c)}{\leq} (1 + r_d(\bar{n}_d)) \|h - f\|_I + r_d(\bar{n}_d) d_{\bar{I}_{\bar{n}_d}}(f, \hat{f}_{\bar{n}_d}) \\
&\stackrel{(d)}{\leq} (1 + r_d(\bar{n}_d)) \|h - f\|_I + r_d(\bar{n}_d) \sum_{J \in \bar{I}_{\bar{n}_d}} |P_J - q_J|.
\end{aligned}
$$

where $(a)$ follows since $(h - \hat{f}_{I, \bar{n}_d}) \in \mathcal{P}_d$, and from definitions of the ratio $r_d(\bar{n}_d)$ in Equation (1), $(b)$ follows since the $d_{\bar{I}}$-distance satisfies the triangle inequality, $(c)$ follows since the $\ell_1$ distance upper bounds $d_{\bar{I}}$−distance, $(d)$ follows since $\hat{f}_{\bar{n}_d}$, by definition, is the polynomial such that $\int_J \hat{f}_{\bar{n}_d} = q_J$ $\forall J \in \bar{I}_{\bar{n}_d}$, and the interval probability $P_J \stackrel{\text{def}}{=} \int_J f$.

Since $P_J \sim \text{Beta}(q_j n, (1 - P_J)n)$, it follows that

$$
\begin{aligned}
\mathbb{E}\|\hat{f}_{\bar{n}_d} - f\|_I &\leq (1 - r_d(\bar{n}_d))\|h - f\|_I + r_d(\bar{n}_d) \sum_{J \in \bar{I}_{\bar{n}_d}} \mathbb{E}|P_J - q_J| \\
&\stackrel{(a)}{\leq} (1 - r_d(\bar{n}_d))\|h - f\|_I + r_d(\bar{n}_d) \sum_{J \in \bar{I}_{\bar{n}_d}} \sqrt{\frac{2q_J}{\pi n}} \\
&\stackrel{(b)}{\leq} (1 + r_d(\bar{n}_d)) \|h - f\|_I + r_d(\bar{n}_d) \sqrt{\frac{2(d+1)q_I}{\pi n}}.
\end{aligned}
$$

where $(a)$ follows from the mean absolute deviation of the Beta distribution, and $(b)$ follows since by the concavity of $\sqrt{x}$ for $x \geq 0$, the sum $\sum_{J \in \bar{I}_{\bar{n}_d}} \sqrt{q_J}$ is maximized if for each $J \in \bar{I}_{\bar{n}_d}$, $q_J = (\sum_{J \in \bar{I}_{\bar{n}_d}} q_J)/|\bar{I}_{\bar{n}_d}| = q_I/(d+1)$.

The following version will be useful in the proof of Lemma 10. Under event $\mathcal{Q}_\epsilon$,

$$
\begin{aligned}
\mathbb{E}\|\hat{f}_{\bar{n}_d} - f\|_I &\leq (1 - r_d(\bar{n}_d))\|h - f\|_I + r_d(\bar{n}_d) \sum_{J \in \bar{I}_{\bar{n}_d}} \mathbb{E}|P_J - q_J| \\
&\leq (1 + r_d(\bar{n}_d)) \|h - f\|_I + r_d(\bar{n}_d) \sum_{J \in \bar{I}_{\bar{n}_d}} \epsilon\sqrt{q_J} \\
&\stackrel{(a)}{\leq} (1 + r_d(\bar{n}_d)) \|h - f\|_I + r_d(\bar{n}_d)\epsilon\sqrt{(d+1)q_I},
\end{aligned}
$$

where $(a)$ follows due to the same reasoning as above.

### C.2 Proof of Lemma 7

**Proof** Fix $h \in \mathcal{P}_d$. Let $(\beta_1, , \ldots, \beta_{d_0})$ be the roots of $h$ in $[0, 1]$ for some $\beta_1 \leq , \ldots, \leq \beta_{d_0}, 0 \leq d_0 \leq d$. Let $\beta_0 \stackrel{\text{def}}{=} 0, \beta_{d_0+1} \stackrel{\text{def}}{=} 1$. Notice that

$$
\begin{aligned}
\int_0^1 |h| = \sum_{i=1}^{d_0+1} \left| \int_{\beta_{i-1}}^{\beta_i} h \right| &\stackrel{(a)}{\leq} \sup_{\bar{m}_d \in \mathcal{N}_d} \sum_{i=1}^{d+1} \left| \int_{m_{i-1}}^{m_i} h \right| \\
&= \sup_{\bar{m}_d \in \mathcal{N}_d} \max_{\bar{s} \in \{0,1\}^{d+1}} \sum_{i=1}^{d+1} (-1)^{s_i} \int_{m_{i-1}}^{m_i} h \\
&\stackrel{(b)}{\leq} \int_0^1 |h|,
\end{aligned}
$$

where $(a)$ follows since on padding $d - d_0$ zeros, $(0, \cdots, 0, \beta_0, \ldots, \beta_{d+1}) \in \mathcal{N}_d$. Thus $(b)$ is, in fact, an equality, implying

$$r_d(\bar{n}_d) = \sup_{h \in \mathcal{P}_d} r(\bar{n}_d, h) = \sup_{h \in \mathcal{P}_d} \frac{\int_0^1 |h|}{\sum_{i=1}^{d+1} |\int_{n_{i-1}}^{n_i} h|}$$

$$= \sup_{h \in \mathcal{P}_d} \frac{\sup_{\bar{m}_d \in \mathcal{N}_d} \max_{\bar{s} \in \{0,1\}^{d+1}} \sum_{i=1}^{d+1} (-1)^{s_i} \int_{m_{i-1}}^{m_i} h}{\sum_{i=1}^{d+1} |\int_{n_{i-1}}^{n_i} h|}$$

$$= \sup_{\bar{m}_d \in \mathcal{N}_d} \max_{\bar{s} \in \{0,1\}^{d+1}} \sup_{h \in \mathcal{P}_d} \frac{\sum_{i=1}^{d+1} (-1)^{s_i} \int_{m_{i-1}}^{m_i} h}{\sum_{i=1}^{d+1} |\int_{n_{i-1}}^{n_i} h|}.$$

Denote $h = \sum_{i=1}^{d+1} c_i \cdot x^{i-1}$ and let $\bar{c} \stackrel{\text{def}}{=} (c_1, \ldots, c_{d+1})$. Notice that since $r(\bar{n}_d, h) \geq 1$ for any $h \in \mathcal{P}_d$, and since $r_d(\bar{n}_d, 0) \stackrel{\text{def}}{=} 1$, WLOG assume $h \neq 0$ or $\bar{c} \neq \bar{0} \stackrel{\text{def}}{=} (0, \ldots, 0)$. By linearity of the integral of $h$ in $\bar{c}$, recast $r_d(\bar{n}_d)$ into

$$r_d(\bar{n}_d) = \sup_{\bar{m}_d \in \mathcal{N}_d} \max_{\bar{s} \in \{0,1\}^{d+1}} \sup_{\bar{c} \in \mathbb{R}^{d+1} \setminus \{\bar{0}\}} \frac{\sum_{i=1}^{d+1} c_i \mu_i}{\sum_{i=1}^{d+1} |\sum_{j=1}^{d+1} c_j \lambda_{i,j}|},$$

where for any $i, j \in \{1, \ldots, d+1\}$, $\mu_i \in \mathbb{R}$ is a function of $\bar{m}_d, \bar{s}$ and $\lambda_{i,j} \in \mathbb{R}$ is a function of $\bar{n}_d$. Observe that $\bar{n}_d$ is given, and additionally fix $\bar{m}_d \in \mathcal{N}_d$, $\bar{s} \in \{0,1\}^{d+1}$. Since the objective function here is a ratio whose denominator is positive (since $h \neq 0$), WLOG set the numerator to 1 via the constraint $\sum_{i=1}^{d+1} c_i \mu_i = 1$ and convert it to a linear program as:

$$\max \frac{1}{\sum_{i=1}^{d+1} v_i} : \bar{c}, \bar{v} \in \mathbb{R}^{d+1}, \quad v_i \geq \sum_{j=1}^{d+1} c_j \lambda_{i,j}, \; v_i \geq -\sum_{j=1}^{d+1} c_j \lambda_{i,j}, \sum_{i=1}^{d+1} c_i \mu_i = 1,$$

where $\bar{v} \stackrel{\text{def}}{=} (v_1, \ldots, v_{d+1})$. Observe that these constraints give rise to a bounded region, and since this is a linear program, there exists a solution at some corner point involving at least $2 \cdot (d+1)$ equalities, one for each variable. In any such solution, since the equality: $\sum_{i=1}^{d+1} c_i \mu_i = 1$ is always active, at least $2 \cdot (d+1) - 1$ of the other inequalities attain equality. Notice that for any $i \in \{1, \ldots, d+1\}$, $v_i = 0$ if both

$$v_i = \sum_{j=1}^{d+1} c_j \lambda_{i,j} \text{ and } v_i = -\sum_{j=1}^{d+1} c_j \lambda_{i,j} \text{ hold.}$$

Thus in this corner point solution, $v_i \neq 0$ for at most one $i \in \{1, \ldots, d+1\}$. Let

$$\mathcal{D}_{\bar{n}_d} = \left\{ \bar{c} \in \mathbb{R}^{d+1} \setminus \{\bar{0}\} : \exists i_1 \in \{1, \ldots, d+1\} : \forall i \neq i_1, |\sum_{j=1}^{d+1} c_j \lambda_{i,j}| = 0 \right\}$$

This implies

$$r_d(\bar{n}_d) = \sup_{\bar{m}_d \in \mathcal{N}_d} \max_{\bar{s} \in \{0,1\}^{d+1}} \sup_{\bar{c} \in \mathbb{R}^{d+1} \setminus \{\bar{0}\}} \frac{\sum_{i=1}^{d+1} c_i \mu_i}{\sum_{i=1}^{d+1} |\sum_{j=1}^{d+1} c_j \lambda_{i,j}|}$$

$$= \sup_{\bar{m}_d \in \mathcal{N}_d} \max_{\bar{s} \in \{0,1\}^{d+1}} \max_{\bar{c} \in \mathcal{D}_{\bar{n}_d}} \frac{\sum_{i=1}^{d+1} c_i \mu_i}{\sum_{i=1}^{d+1} |\sum_{j=1}^{d+1} c_j \lambda_{i,j}|}$$

$$= \sup_{\bar{m}_d \in \mathcal{N}_d} \max_{\bar{s} \in \{0,1\}^{d+1}} \max_{h \in \mathcal{P}_{\bar{n}_d}} \frac{\sum_{i=1}^{d+1} (-1)^{s_i} \int_{m_{i-1}}^{m_i} h}{\sum_{i=1}^{d+1} |\int_{n_{i-1}}^{n_i} h|}$$

$$= \max_{h \in \mathcal{P}_{\bar{n}_d}} \sup_{\bar{m}_d \in \mathcal{N}_d} \max_{\bar{s} \in \{0,1\}^{d+1}} \frac{\sum_{i=1}^{d+1} (-1)^{s_i} \int_{m_{i-1}}^{m_i} h}{\sum_{i=1}^{d+1} |\int_{n_{i-1}}^{n_i} h|}$$

$$= \max_{h \in \mathcal{P}_{\bar{n}_d}} \frac{\int_0^1 |h|}{\sum_{i=1}^{d+1} |\int_{n_{i-1}}^{n_i} h|} = \max_{h \in \mathcal{P}_{\bar{n}_d}} r_d(\bar{n}_d, h).$$

### C.3 Proof of Lemma 8

**Proof** For any polynomial $h \in \mathcal{P}_d$, the ratio $r(\bar{n}_d, h)$ is invariant to multiplying both the numerator and denominator by a constant. Thus, WLOG consider polynomials whose leading coefficient is 1. Then for any $\bar{n}_d \in \mathcal{N}_d$, $\mathcal{P}_{\bar{n}_d} = (h_{\bar{n}_d,1}, \ldots, h_{\bar{n}_d,d+1})$, is a set consisting of $d+1$ unique polynomials, where each $h_{\bar{n}_d,i}$, $i \in \{1, \ldots, d+1\}$ is that polynomial with 0 area in all intervals in $\bar{I}_{\bar{n}_d}$ except $I_{\bar{n}_d,i}$.

**Case d = 0:** Here $\mathcal{N}_0 = \{(0,1)\}$ and is a singleton set. Since any $h \in \mathcal{P}_0$ is a constant value, $\int_0^1 |h| = |\int_0^1 h|$. Therefore $r_0^\star = \max_{h \in \mathcal{P}_0} r(\bar{n}_d, h) = 1$.

**Case d = 1:** Let $\bar{n}_1 = (0, m, 1)$. In this case $h_{\bar{n}_d,1}(x) = x - m/2$ and $h_{\bar{n}_d,2}(x) = x - (1+m)/2$. Using Lemma 7,

$$r_1(\bar{n}_d) = \max_{h \in \mathcal{P}_{\bar{n}_d}} r(\bar{n}_d, h) = \max\{r(\bar{n}_d, h_{\bar{n}_d,1}), r(\bar{n}_d, h_{\bar{n}_d,2})\}$$

$$= \max \left\{ \frac{m^2/4 + (1-m/2)^2}{1-m}, \frac{(1-m)^2/4 + ((1+m)/2)^2}{m} \right\}.$$

$r_1(\bar{n}_d)$ is minimized for $m^\star = 1/2$, giving $r_1^\star = (1/16 + 9/16)/(1/2) = 1.25$.

**Case d = 2:** By symmetry, the minimizing node partition is symmetric about 0.5. Thus WLOG let $\bar{n}_2 = (0, m, 1-m, 1)$ for some $m \leq 0.5$. Among the $d+1 = 3$ polynomials in $\mathcal{P}_{\bar{n}_2}$, by symmetry of $\bar{n}_2$, $r_2(h_{\bar{n}_d,1}) = r_2(h_{\bar{n}_d,3})$. Thus we consider the larger ratio across only two polynomials, $h_{\bar{n}_d,2}, h_{\bar{n}_d,3}$.

Denote the polynomial as $h_{\bar{n}_d,2}(x) = (x-a_2)^2 - b_2^2$ and upon setting the respective integrals to 0,

$$\left| \frac{m^3}{3} - a_2 m^2 + (a_2^2 - b_2^2)m \right| = 0, \quad \left| \frac{1-(1-m)^3}{3} - a_2(1-(1-m)^2) + (a_2^2 - b_2^2)m \right| = 0$$

$$\implies a_2 = \frac{1}{2}, b_2^2 = \frac{3(m^2-m)+1}{9}.$$

Representing $h_{\bar{n}_d,3}(x) = (x-a_3)^2 - b_3^2$ and repeating the same steps,

$$\left| \frac{m^3}{3} - a_3 m^2 + (a_3^2 - b_3^2)m \right| = 0, \quad \left| \frac{(1-m)^3 - m^3}{3} - a_3((1-m)^2 - m^2) + (a_3^2 - b_3^2)(1-2m) \right| = 0$$

$$\implies a_3 = \frac{1}{3}, b_3^2 = \frac{4m^2 - 6m + 3}{3}.$$

The corresponding $r(\bar{n}_d, h_{\bar{n}_d,2})$ and $r(\bar{n}_d, h_{\bar{n}_d,3})$ are given by

$$\frac{8 \left( \frac{1-3m(1-m)}{9} \right)^{3/2}}{m(1-m)} + 1, \quad \frac{2 \left( \frac{(2m-1)(2m-2)+1}{3} \right)^{3/2}}{(2m-1)(m-1)} - 1.$$

From simultaneously minimizing the above expressions by equating them, the optimal $m$ is the root of

$$q_2(m) = -\frac{26624}{729}m^{14} + \frac{193280}{729}m^{13} - \frac{211024}{243}m^{12} + \frac{3703648}{2187}m^{11} - \frac{4790776}{2187}m^{10}$$
$$+ \frac{39108232}{19683}m^9 - \frac{8554775}{6561}m^8 + \frac{12357280}{19683}m^7 - \frac{13004032}{59049}m^6 + \frac{1061792}{19683}m^5$$
$$- \frac{4350752}{531441}m^4 + \frac{246976}{531441}m^3 + \frac{11840}{177147}m^2 - \frac{6656}{531441}m + \frac{256}{531441}$$

near 0.26. Thus the optimal $m^\star \approx 0.2599$ and the corresponding $r_2^\star \approx 1.423$.

**Case d = 3:** By symmetry, as before, WLOG let $\bar{n}_d = (0, m, 0.5, 1-m, 1)$. This reduces the search space to just two polynomials, $h_{\bar{n}_d,1}, h_{\bar{n}_d,2}$. The optimal $m$ occurs as the root of

$$q_3(m) \stackrel{\text{def}}{=} m^{69} + \frac{2233}{46}m^{68} + \frac{3394851}{2944}m^{67} - \frac{26295551}{1472}m^{66} + \frac{76466381715}{376832}m^{65}$$

$$- \frac{1357944230009}{753664}m^{64} + \frac{627961733592749}{48234496}m^{63} - \frac{3795194179761079}{48234496}m^{62}$$

$$+ \frac{1252499739594399621}{3087007744}m^{61} - \frac{5593584650474780121}{3087007744}m^{60}$$

$$+ \frac{87541700408454835933}{12348030976}m^{59} - \frac{9689649149944354300097}{395136991232}m^{58}$$

$$+ \frac{47748138828085878102175}{6322191859712}m^{57} - \frac{131681673633637796401265}{6322191859712}m^{56}$$

$$+ \frac{1045188536455255357264264 11}{202310139510784}m^{55} - \frac{9357299571916607313304 80575}{809240558043136}m^{54}$$

$$+ \frac{18940320032160659182562501 47}{809240558043136}m^{53} - \frac{11107006368666590512165725 2873}{25895697857380352}m^{52}$$

$$+ \frac{294796693788038263639833772 3253}{414331165718085632}m^{51} - \frac{9605313015982677914847282 6511}{9007199254740992}m^{50}$$

$$+ \frac{595776877329189835594414388 1565}{414331165718085632}m^{49}$$

$$- \frac{286660928003091885687566677 23285}{1657324662872342528}m^{48}$$

$$+ \frac{193682525931014771367725908 7614429}{106068778423829921792}m^{47}$$

$$- \frac{216663456959495677102483903 955187}{13258597302978740224}m^{46}$$

$$+ \frac{479768993444696163016003164 3189779}{424275113695319687168}m^{45}$$

$$- \frac{660991760338697881312887281 5736861}{1697100454781278748672}m^{44}$$

$$- \frac{183105129495273434934912422 9564767}{424275113695319687168}m^{43}$$

$$+ \frac{774482572492759626728073840 94624197}{67884018191125114994688}m^{42}$$

$$- \frac{343821157459641486443539964 8557575571}{217228858212003679830016}m^{41}$$

$$+ \frac{146918733414150435554179611 21466375911}{868915432848014719320064}m^{40}$$

$$- \frac{521970584129282136669302330 26164438477}{34756617313392058877280256}m^{39}$$

$$+ \frac{626588406181557032617836659 688444683449}{5561058770227294203648409 6}m^{38}$$

$$- \frac{155552505150918877198073027 8198868813547}{2224423508090917681459363 84}m^{37}$$

$$+ \frac{291172834873853037098695003 9544396475929}{88976940323636707258374553 6}m^{36}$$

$$- \frac{415629763783269606797247480 824967018319}{6189700196426901374495621 12}m^{35}$$

$$- \frac{212939502278553250962033810 76755307029285}{28472620903563746322679857 152}m^{34}$$

$$+ \frac{568117182843420536224530134 72967651433283}{45556193445701994116287771 4432}m^{33}$$

$$- \frac{544176155073471826876603760 623685652791461}{45556193445701994116287771 4432}m^{32}$$

$$+ \frac{265107922318232071114682080 52472537291091795}{29155963805249276234424173 723648}m^{31}$$

$$- \frac{433925296736104935336803334 22356143484711}{72889909513123190586060434 30912}m^{30}$$

$$+\frac{80732188658524254663695919868749449919506951}{2332477104419942098753933897891 84}m^{29}$$

$$-\frac{21178612540977853104278860543475186392271681}{11662385522099710493769669489 4592}m^{28}$$

$$+\frac{80945977995703186772569013525777965350114711}{9329908417679768395015735591 56736}m^{27}$$

$$-\frac{17702912107607481296775724392303230443070095}{46649542088398841975078677957 9578368}m^{26}$$

$$+\frac{11379568962839820770933004526627499941 0341749}{7463926734143814716012588473253888}m^{25}$$

$$-\frac{21034631455622622763256522489198166970683613}{373196336707190735800629423662 6944}m^{24}$$

$$+\frac{22916753247501379766927497305407136259487 2031}{11942282774630103545620141557206220 8}m^{23}$$

$$-\frac{17963282649430914596699742603270996080670419}{2985570693657525886405035389301 5552}m^{22}$$

$$+\frac{16585753285425745935965124906360351743012 7219}{9553826219704082836496113245764976 64}m^{21}$$

$$-\frac{21988393007280087042133518345726499678514793}{4776913109852041182480566228824 8832}m^{20}$$

$$+\frac{26730625170682089529622024014578870916 20815}{2388456554926020709124028311441244 16}m^{19}$$

$$-\frac{47535737602485273668594974352674534173 84687}{19107652439408165672992226491529953 28}m^{18}$$

$$+\frac{76985304326121722168401741264209173943 35697}{15286121951526532538393781193223962 624}m^{17}$$

$$-\frac{88254391793832097376072061503233796540539}{9553826219704082836496113245764976 64}m^{16}$$

$$+\frac{46618175901212180036351783955592685629 2931}{30572243903053065076787562386447925 248}m^{15}$$

$$-\frac{34314489379863699139383468530966229926169}{15286121951526532538393781193223962 624}m^{14}$$

$$+\frac{71176987410160907949890583502121075660389}{24457795122442452061430049909158340 1984}m^{13}$$

$$-\frac{3987333962073343668163889901656755306225}{12228897561221226030715024954579170 0992}m^{12}$$

$$+\frac{74866865511474035460074506627157026503 1}{24457795122442452061430049909158340198 4}m^{11}$$

$$+\frac{1108452567613061636064406694422926370 27}{48915590244884904122860099818316680 3968}m^{10}$$

$$+\frac{18991769640838597032830448297436332 09}{170141183460469231731687303715884 105728}m^{9}$$

$$-\frac{2073921584792354563737120211683341}{42535295865117307932921825928971 026432}m^{8}$$

$$-\frac{34402390201821002633790825125211 75}{611444878061061301535751247728958 50496}m^{7}$$

$$+\frac{4264138856148752641548430451717}{664613997892457936451903530140 172288}m^{6}$$

$$-\frac{3117503551035781118929644883731}{764306097576326626919689059661 1981312}m^{5}$$

$$+\frac{290370771820371191127221254 23}{191076524394081656729922264 9152995328}m^{4}$$

$$-\frac{460833093614235731783726 79}{2388456554926020709124028 311441244 16}m^{3}$$

$$- \frac{59781192731840360565685541}{5971141387315051772810070778 6031104} m^2$$
$$+ \frac{30104861649982869480831}{5971141387315051772810070778 6031104} m$$
$$- \frac{10590565578289797 6459}{14927853468287629432025176946507776}$$

near 0.155. This gives $m^\star \approx 0.1548$ and the corresponding $r_3^\star \approx 1.559$.

For degrees $4 \leq d \leq 8$, we use numerical methods on top of Lemma 7 to derive $\bar{n}_d \in \mathcal{N}_d$ and the corresponding $r_d(\bar{n}_d)$. These values populate the second table in Lemma 8.

# D   The Compare and Stitch Routines

## D.1   Proof of Lemma 9

**Proof**   As observed in Equation (**??**), any $q \in \bar{q} \in \Delta_{\text{emp,n}}$ is an integral multiple of $1/n$. Observing that $\log_2 n$ is an integer, we may decompose $q$ along its binary expansion as

$$q = \sum_{j=0}^{\log_2 n} 2^{-j} b_j,$$

for some $b_j \in \{0, 1\}$, $j \in \{1, \ldots, \log_2 n\}$. Replace each $q \in \bar{q}$ with the vector $(2^{-0} b_0, 2^{-1} b_1, \cdots)$ to obtain $\bar{q}' \in \Delta_{\text{bin,n}}$. From the property of the geometric sum,

$$\sum_{j=0}^{\log_2 n} \sqrt{2^{-j} b_j} \leq \frac{\sqrt{q}}{\sqrt{2} - 1}.$$

Finally $\|f^\star_{\bar{I}_{\bar{q}'}} - f\|_1 \leq \|f^\star_{\bar{I}_{\bar{q}}} - f\|_1$ since $\bar{I}_{\bar{q}'}$ being a finer partition than $\bar{I}_{\bar{q}}$, $f^\star_{\bar{I}_{\bar{q}'}}$ is a closer approximation to $f$ than $f^\star_{\bar{I}_{\bar{q}}}$.

## D.2   Proof of Lemma 10

**Proof**   For any interval $I$, let

$$f^\star_I \overset{\text{def}}{=} \arg \min_{h \in \mathcal{P}_d} \|h - f\|_I,$$

and for any partition $\bar{I}$, let $f^\star_{\bar{I}}$ be the piecewise polynomial that equals $f^\star_I$ in each $I \in \bar{I}$. For simplicity let $I_{\bar{q}} \overset{\text{def}}{=} I_{\bar{q}_D}$ denote the final partition and $\bar{q} \overset{\text{def}}{=} \bar{q}_D$ the corresponding empirical distribution. Consider any $\bar{p} \in \Delta_{\text{bin,n}}$ and its associated interval partition, $\bar{I}_{\bar{p}}$. Two interval partitions $\bar{I}_1, \bar{I}_2$ corresponding to binary distributions have the following property: Any interval in $\bar{I}_1$ is either completely contained within some interval in $\bar{I}_2$, or is a union of contiguous intervals from $\bar{I}_2$. As a result $\bar{I}_{\bar{q}}$ may partitioned into three classes of intervals:

Figure 5: Illustration of $\bar{I}_{\bar{q}}$ being partitioned into $\bar{I}^1_{\bar{q}}$, $\bar{I}^2_{\bar{q}}$ and $\bar{I}^3_{\bar{q}}$ using $\bar{I}_{\bar{p}}$.

- $\bar{I}^1_{\bar{q}}$, composed of intervals that are equal to some interval in $\bar{I}_{\bar{p}}$,

- $\bar{I}^2_{\bar{q}}$, that consists of intervals that lie strictly within some interval in $\bar{I}_{\bar{p}}$,

- $\bar{I}^3_{\bar{q}}$, containing intervals that are unions of more than one interval from $\bar{I}_{\bar{p}}$.

This is shown in Figure 5. Lemmas 11, 12, 13 address each of these intervals separately. Combining the lemmas,

$$
\begin{aligned}
\|\hat{f}_{\bar{I}_{\bar{q}}} - f\|_1 &= \|\hat{f}_{\bar{I}_{\bar{q}}} - f\|_{\bar{I}_{\bar{q}}^1} + \|\hat{f}_{\bar{I}_{\bar{q}}} - f\|_{\bar{I}_{\bar{q}}^2} + \|\hat{f}_{\bar{I}_{\bar{q}}} - f\|_{\bar{I}_{\bar{q}}^3} \\
&\leq (r_d + 1) \cdot \|f^\star_{\bar{I}_{\bar{p}}} - f\|_{\bar{I}_{\bar{q}}^1} + \sum_{I \in \bar{I}_{\bar{p}}^1} r_d \cdot \epsilon \sqrt{(d+1)p_I} \\
&\quad + \frac{(r_d + 1) \cdot \alpha}{\alpha - 2} \cdot \|f^\star_{\bar{I}_{\bar{p}}} - f\|_{\bar{I}_{\bar{q}}^2} + \frac{1}{\alpha - 1} \sum_{I \in \bar{I}_{\bar{p}}^2} r_d \cdot \epsilon \sqrt{(d+1)p_I} \\
&\quad + (r_d + 1) \cdot \|f^\star_{\bar{I}_{\bar{p}}} - f\|_{\bar{I}_{\bar{q}}^3} + \frac{\alpha\sqrt{2} + \sqrt{2} - 1}{\sqrt{2} - 1} \sum_{I \in \bar{I}_{\bar{p}}^3} r_d \cdot \epsilon \sqrt{(d+1)p_I} \\
&\overset{(a)}{\leq} \frac{(r_d + 1) \cdot \alpha}{\alpha - 2} \|f^\star_{\bar{I}_{\bar{p}}} - f\|_1 + \frac{\alpha\sqrt{2} + \sqrt{2} - 1}{\sqrt{2} - 1} \sum_{I \in \bar{I}_{\bar{p}}} r_d \cdot \epsilon \sqrt{(d+1)p_I},
\end{aligned}
$$

where $(a)$ follows since $\alpha > 2 \Rightarrow 1/(\alpha - 1) < 1 < (\alpha\sqrt{2} + \sqrt{2} - 1)/(\sqrt{2} - 1)$.

**Lemma 11.** *For the final partition $\bar{I}_{\bar{q}}$ in the run of* MERGE *and any $\bar{p} \in \Delta_{\text{bin,n}}$, let $\bar{I}_{\bar{q}}^1 \subseteq \bar{I}_{\bar{q}}$ be the intervals that intersect with $\bar{I}_{\bar{p}}$. Let $\bar{I}_{\bar{p}}^1 = \bar{I}_{\bar{q}}^1 \subseteq \bar{I}_{\bar{p}}$ denote the corresponding collection in $\bar{I}_{\bar{p}}$. Then,*

$$
\|\hat{f}_{\bar{I}_{\bar{q}}} - f\|_{\bar{I}_{\bar{q}}^1} \leq (r_d + 1) \cdot \|f^\star_{\bar{I}_{\bar{p}}} - f\|_{\bar{I}_{\bar{q}}^1} + \sum_{I \in \bar{I}_{\bar{p}}^1} r_d \cdot \epsilon \sqrt{(d+1)p_I},
$$

**Proof**  Follows from Theorem 1 and noticing that intervals in $\bar{I}_{\bar{q}}^1$ and $\bar{I}_{\bar{p}}^1$ coincide.

**Lemma 12.** *For the final partition $\bar{I}_{\bar{q}}$ in the run of* MERGE *and any $\bar{p} \in \Delta_{\text{bin,n}}$, let $\bar{I}_{\bar{q}}^2 \subseteq \bar{I}_{\bar{q}}$ be the intervals that do not intersect with, and strictly lie in some interval in $\bar{I}_{\bar{p}}$. Let $\bar{I}_{\bar{p}}^2 \subseteq \bar{I}_{\bar{p}}$ be the corresponding intervals that contain $\bar{I}_{\bar{q}}^2$. Then,*

$$
\|\hat{f}_{\bar{I}_{\bar{q}}} - f\|_{\bar{I}_{\bar{q}}^2} \leq \frac{(r_d + 1) \cdot \alpha}{\alpha - 2} \cdot \|f^\star_{\bar{I}_{\bar{p}}} - f\|_{\bar{I}_{\bar{q}}^2} + \frac{1}{\alpha - 1} \sum_{I \in \bar{I}_{\bar{p}}^2} r_d \cdot \epsilon \sqrt{(d+1)p_I}.
$$

**Proof**  Notice that all intervals in $\bar{I}_{\bar{q}}^2$ are strictly contained within some interval in $\bar{I}_{\bar{p}}^2$. Using this, we further partition $\bar{I}_{\bar{q}}^2$ using intervals in $\bar{I}_{\bar{p}}^2$. Fix an $I \in \bar{I}_{\bar{p}}^2$ and let $\bar{I} \in \bar{I}_{\bar{q}}^2$ be intervals whose union gives $I$. Let $\bar{q}_{\bar{I}} \subseteq \bar{q}$ denote the empirical probabilities corresponding to $\bar{I}$ and let $p_I$ denote the empirical probability under $I$.

Figure 6: Illustration of $\bar{I} \in \bar{I}_{\bar{q}}^2$, $\bar{I}_{\bar{s}_I}$ and $\bar{I}_{\bar{s}_I^1}$ corresponding to a particular $I \in \bar{I}_{\bar{p}}^2$.

While $\bar{q}_{\bar{I}}$ is a sub-distribution in general, WLOG assume $\bar{q}_{\bar{I}}$ is a distribution. Now, at some point in the run of MERGE, COMP was called with $\hat{f}_{I,\text{INT}}$, $\bar{I}$, $\bar{q}_{\bar{I}}$, and it was in-turn declared that $\bar{I}$ was not to be merged into $I$. Therefore, for the $\mu_{\bar{I},\gamma}(\hat{f}_{I,\text{INT}})$ attaining binary distribution, $\bar{s}_I \in \Delta_{\text{bin,n},\geq \bar{q}_{\bar{I}}}$,

$\Lambda_{I_{\bar{s}_I}}(\hat{f}_{I,\text{INT}}) - \lambda_{\bar{s}_I,\gamma} \geq 0$. It follows that

$$\sum_{s \in \bar{s}_I} \alpha \cdot r_d \cdot \epsilon \sqrt{(d+1)s} = \lambda_{\bar{s}_I,\gamma} \leq \Lambda_{I_{\bar{s}_I}}(\hat{f}_{I,\text{INT}})$$

$$= \|\hat{f}_{I,\text{INT}} - \hat{f}_{\bar{I}_{\bar{s}_I}}\|_I$$

$$\leq \|\hat{f}_{I,\text{INT}} - f\|_I + \|f - \hat{f}_{\bar{I}_{\bar{s}_I}}\|_I$$

$$\overset{(a)}{\leq} (r_d + 1) \cdot \|f_I^\star - f\|_I + r_d \cdot \epsilon\sqrt{(d+1)p_I}$$

$$+ (r_d + 1) \cdot \|f_{\bar{I}_{\bar{s}_I}}^\star - f\|_I + \sum_{s \in \bar{s}_I} r_d \cdot \epsilon\sqrt{(d+1)s}$$

$$\overset{(b)}{\leq} 2(r_d + 1) \cdot \|f_I^\star - f\|_I + r_d \cdot \epsilon\sqrt{(d+1)p_I} + \sum_{s \in \bar{s}_I} r_d \cdot \epsilon\sqrt{(d+1)s},$$

where $(a)$ follows from in Theorem 1, $(b)$ follows since $I$ being the union of $\bar{I}_{\bar{q}_I}$ is also the union of $\bar{I}_{\bar{s}_I}$, and $f_I^\star$ is therefore a coarser approximation to $f$ than $f_{\bar{I}_{\bar{s}_I}}^\star$, giving rise to a larger $\ell_1$ distance. Rearrange this to obtain

$$\sum_{s \in \bar{s}_I} r_d \cdot \epsilon\sqrt{(d+1)s} \leq \frac{1}{\alpha - 1} \cdot \left(2(r_d + 1) \cdot \|f_I^\star - f\|_I + r_d \cdot \epsilon\sqrt{(d+1)p_I}\right). \tag{5}$$

Consider a fixed $I' \in \bar{I}_{\bar{s}_I}$ and let $\bar{I}' \in \bar{I}$ be the intervals under $\bar{I}$ whose union gives $I'$. We recursively use the same argument to bound the LHS of Equation (5). This is shown for the leftmost interval of $\bar{I}_{\bar{s}_I}$ in Figure 6. Let $\bar{q}_{\bar{I}'}$ be the corresponding probabilities under $\bar{I}'$ and let $s_{I'}$ denote the empirical probability under $I'$. Notice that in some previous step of MERGE, as was for $I$, COMP was invoked with $\hat{f}_{I',\text{INT}}$, $\bar{I}'$, $\bar{q}_{\bar{I}'}$, for which $\mu_{\bar{I}',\gamma}(\hat{f}_{I',\text{INT}}) \geq 0$. Repeat the same procedure as above to obtain

$$\sum_{s \in \bar{s}_{I'}} r_d \cdot \epsilon\sqrt{(d+1)s} \leq \frac{1}{\alpha - 1} \cdot \left(2(r_d + 1) \cdot \|f_{I'}^\star - f\|_{I'} + r_d \cdot \epsilon\sqrt{(d+1)s_{I'}}\right)$$

$$\overset{(a)}{\leq} \frac{1}{\alpha - 1} \cdot \left(2(r_d + 1) \cdot \|f_I^\star - f\|_{I'} + r_d \cdot \epsilon\sqrt{(d+1)s_{I'}}\right), \tag{6}$$

where $\bar{s}_{I'}$ here is the binary distribution which attains $\mu_{\bar{I}',\gamma}(\hat{f}_{I',\text{INT}})$, and $(a)$ follows because $I'$ being an interval within $I$, $f_I^\star$ is a coarser approximation to $f$ than $f_{I'}^\star$. Summing Equation (6) for each such $I'$, accumulate the distribution $\bar{s}_I^1 \overset{\text{def}}{=} (\cup_{I' \in \bar{I}_{\bar{s}_I}} \bar{s}_{I'})$, and using Equation (5), the inequality,

$$\sum_{s \in \bar{s}_I^1} r_d \cdot \epsilon\sqrt{(d+1)s} \leq \left(\frac{1}{\alpha - 1} + \frac{1}{(\alpha-1)^2}\right) \cdot 2(r_d+1) \cdot \|f_I^\star - f\|_I + \frac{1}{\alpha-1} \cdot r_d \cdot \epsilon\sqrt{(d+1)p_I}. \tag{7}$$

Notice that while both $\bar{s}_I, \bar{s}_I^1 \in \Delta_{\text{bin,n},\geq \bar{q}_I}$, $\bar{s}_I^1$ is at least one notch closer to $\bar{q}_I$ as $\bar{s}_I^1 \in \Delta_{\text{bin,n},<\bar{s}_I}$. Since the number of binary distributions is finite, on recursively using this argument, summation across $\bar{q}_I$ is eventually obtained on the LHS. Iterating on this procedure yields the upper bound

$$\sum_{q \in \bar{q}_I} r_d \cdot \epsilon\sqrt{(d+1)q} \leq \left(\frac{1}{\alpha - 1} + \frac{1}{(\alpha-1)^2} + \cdots\right) \cdot 2(r_d + 1) \cdot \|f_I^\star - f\|_I$$

$$+ \frac{1}{\alpha-1} r_d \cdot \epsilon\sqrt{(d+1)p_I}$$

$$\overset{(a)}{\leq} \frac{2(r_d + 1)}{\alpha - 2} \cdot \|f_I^\star - f\|_I + \frac{1}{\alpha - 1} \cdot r_d \cdot \epsilon\sqrt{(d+1)p_I},$$

where $(a)$ follows since $\alpha > 2$. Repeating this argument across each $I \in \bar{I}_{\bar{p}}^2$,

$$\sum_{I \in \bar{I}_{\bar{q}}^2} r_d \cdot \epsilon\sqrt{(d+1)q_I} \leq \frac{2(r_d + 1)}{\alpha - 2} \cdot \|f_{\bar{I}_{\bar{p}}^2}^\star - f\|_{\bar{I}_{\bar{q}}^2} + \frac{1}{\alpha - 1} \sum_{I \in \bar{I}_{\bar{p}}^2} r_d \cdot \epsilon\sqrt{(d+1)p_I}. \tag{8}$$

This finally gives us

$$\|\hat{f}_{\bar{I}_{\bar{q}}} - f\|_{\bar{I}_{\bar{q}}^2} \overset{(a)}{\leq} (r_d + 1) \cdot \|f^{\star}_{\bar{I}_{\bar{q}}} - f\|_{\bar{I}_{\bar{q}}^2} + \sum_{I \in \bar{I}_{\bar{q}}^2} r_d \cdot \epsilon \sqrt{(d+1)q_I}$$

$$\overset{(b)}{\leq} (r_d + 1) \cdot \|f^{\star}_{\bar{I}_{\bar{p}}} - f\|_{\bar{I}_{\bar{q}}^2} + \sum_{I \in \bar{I}_{\bar{q}}^2} r_d \cdot \epsilon \sqrt{(d+1)q_I}$$

$$\leq (r_d + 1) \left(1 + \frac{2}{\alpha - 2}\right) \|f^{\star}_{\bar{I}_{\bar{p}}} - f\|_{\bar{I}_{\bar{q}}^2} + \frac{1}{\alpha - 1} \sum_{I \in \bar{I}_{\bar{p}}^2} r_d \cdot \epsilon \sqrt{(d+1)p_I},$$

where $(a)$ follows from Theorem 1, $(b)$ follows since, by definition, intervals in $\bar{I}_{\bar{q}}^2$ lie within those in $\bar{I}_{\bar{p}}^2$, and thus $f^{\star}_{\bar{I}_{\bar{p}}}$ is a coarser approximation to $f$ than $f^{\star}_{\bar{I}_{\bar{q}}}$ in $\bar{I}_{\bar{q}}^2$, and finally $(c)$ follows by plugging in Equation (8).

**Lemma 13.** *For the final partition $\bar{I}_{\bar{q}}$ in the run of* MERGE *and any $\bar{p} \in \Delta_{\text{bin,n}}$, let $\bar{I}_{\bar{q}}^3 \subseteq \bar{I}_{\bar{q}}$ be intervals that are unions of more than one interval from $\bar{I}_{\bar{p}}$. Let $\bar{I}_{\bar{p}}^3 \subseteq \bar{I}_{\bar{p}}$ be the corresponding intervals whose union gives $\bar{I}_{\bar{q}}^3$. Then,*

$$\|\hat{f}_{\bar{I}_{\bar{q}}} - f\|_{\bar{I}_{\bar{q}}^3} \leq (r_d + 1) \cdot \|f^{\star}_{\bar{I}_{\bar{p}}} - f\|_{\bar{I}_{\bar{q}}^3} + \frac{\alpha\sqrt{2} + \sqrt{2} - 1}{\sqrt{2} - 1} \sum_{I \in \bar{I}_{\bar{p}}^3} r_d \cdot \epsilon \sqrt{(d+1)p_I}.$$

**Proof**    Fix an $I \in \bar{I}_{\bar{q}}^1$ and let $q_I$ be its empirical probability. Let $\bar{I}_{\bar{p},I} \in \bar{I}_{\bar{p}}$ indicate intervals under $\bar{I}_{\bar{p}}$ whose union gives $I$ and let $\bar{p}_I \subseteq \bar{p}$ denote the corresponding empirical probabilities under $\bar{I}_{\bar{p},I}$. In run of MERGE, let the interval collection that was merged to create $I$ be denoted by $\bar{I}_{\bar{q},I}$, and its collection of empirical probabilities by $\bar{q}_I$. While $\bar{p}_I$ is a sub-distribution in general, WLOG assume it is a distribution. This also implies $\bar{q}_I$ is a distribution.

Using $\bar{I}_{\bar{q},I}$, separate $\bar{I}_{\bar{p},I}$ into

- $\bar{I}_{\bar{p},I}^1$, consisting of intervals in $\bar{I}_{\bar{p},I}$ that are equal to, or unions of intervals from $\bar{I}_{\bar{q},I}$.

- $\bar{I}_{\bar{p},I}^2$, intervals in $\bar{I}_{\bar{p},I}$ that lie strictly inside some interval in $\bar{I}_{\bar{q},I}$.

Let $\bar{I}_{\bar{q},I}^2 \subseteq$ be the corresponding intervals in $\bar{I}_{\bar{q},I}$ that contain $\bar{I}_{\bar{p},I}^2$. Let $\bar{p}_I^1, \bar{p}_I^2$ be empirical probabilities corresponding to $\bar{I}_{\bar{p},I}^1, \bar{I}_{\bar{p},I}^2$ respectively. Similarly let $\bar{q}_I^2$ correspond to $\bar{I}_{\bar{q},I}^2$. This is shown in Figure 7, where the arrow indicates the collection of intervals merged by MERGE.

Figure 7: Illustration of proof construction for a particular $I \in \bar{I}_{\bar{q}}^3$.

Modify $\bar{I}_{\bar{p},I}$ to obtain a new partition $\bar{J}_{\bar{p},I} \stackrel{\text{def}}{=} \bar{I}^1_{\bar{p},I} \cup \bar{I}^2_{\bar{q},I}$. Now each interval in $\bar{J}_{\bar{p},I}$ is equal to, or is a union of intervals from $\bar{I}_{\bar{q},I}$. Equivalently, if $\bar{s}$ is the empirical distribution corresponding to $\bar{J}_{\bar{p},I}$, $\bar{s} \in \Delta_{\text{bin},n,\geq \bar{q}_I}$. Since $I$ was merged when the merging routine was called with $\hat{f}_{I,\text{INT}}, \bar{I}_{\bar{q},I}, \bar{q}_I$, it implies $\lambda_{\bar{s},\gamma} \geq \Lambda_{I_{\bar{s}}}(\hat{f}_{I,\text{INT}})$. Therefore

$$\begin{aligned}
\|\hat{f}_{\bar{I}_{\bar{q}}} - f\|_I &\leq \|\hat{f}_{\bar{J}_{\bar{p},I}} - f\|_I + \|\hat{f}_{\bar{I}_{\bar{q}}} - \hat{f}_{\bar{J}_{\bar{p},I}}\|_I \\
&\stackrel{(a)}{=} \|\hat{f}_{\bar{J}_{\bar{p},I}} - f\|_I + \Lambda_{I_{\bar{s}}}(\hat{f}_{I,\text{INT}}) \\
&\leq \|\hat{f}_{\bar{J}_{\bar{p},I}} - f\|_I + \lambda_{\bar{s},\gamma} \\
&\stackrel{(b)}{=} \|\hat{f}_{\bar{J}_{\bar{p},I}} - f\|_{\bar{J}_{\bar{p},I}} + \lambda_{\bar{s},\gamma} \\
&= \|\hat{f}_{\bar{J}_{\bar{p},I}} - f\|_{\bar{J}_{\bar{p},I}} + \alpha \sum_{s \in \bar{s}} r_d \cdot \epsilon \sqrt{(d+1)s} \\
&\stackrel{(c)}{=} \|\hat{f}_{\bar{I}^1_{\bar{p},I}} - f\|_{\bar{I}^1_{\bar{p},I}} + \alpha \sum_{p \in \bar{p}^1_I} r_d \cdot \epsilon \sqrt{(d+1)p} \\
&\quad + \|\hat{f}_{\bar{I}^2_{\bar{q},I}} - f\|_{\bar{I}^2_{\bar{q},I}} + \alpha \sum_{q \in \bar{q}^2_I} r_d \cdot \epsilon \sqrt{(d+1)q} \\
&\stackrel{(d)}{=} \|\hat{f}_{\bar{I}_{\bar{p}}} - f\|_{\bar{I}^1_{\bar{p},I}} + \alpha \sum_{p \in \bar{p}^1_I} r_d \cdot \epsilon \sqrt{(d+1)p} \\
&\quad + \|\hat{f}_{\bar{I}^2_{\bar{q},I}} - f\|_{\bar{I}^2_{\bar{q},I}} + \alpha \sum_{q \in \bar{q}^2_I} r_d \cdot \epsilon \sqrt{(d+1)q}, \quad (9)
\end{aligned}$$

where $(a)$ follows since by definition, $\hat{f}_{\bar{I}_{\bar{q}}} = \hat{f}_{I,\text{INT}}$ in interval $I$, $(b)$ follows since $\bar{J}_{\bar{p},I}$ being a partition of $I$ lies in the same region as $I$, $(c)$ follows since $\bar{J}_{\bar{p},I} = \bar{I}^1_{\bar{p},I} \cup \bar{I}^2_{\bar{q},I}$, and $(d)$ follows since $\hat{f}_{\bar{I}_{\bar{p}}} = \hat{f}_{\bar{I}^1_{\bar{p},I}}$ in $\bar{I}^1_{\bar{p},I}$ as $\bar{I}^1_{\bar{p},I} \subseteq \bar{I}_{\bar{p}}$.

Now consider an interval $I' \in \bar{I}^2_{\bar{q},I}$. Since $\bar{I}^2_{\bar{q},I} \subseteq \bar{I}_{\bar{q},I}$, and since $\bar{I}_{\bar{q},I}$, by definition, are intervals that were merged to produce $I$, it follows that $I'$ in turn was an interval that was merged into in some previous step of MERGE. As before, let the intervals that were merged to generate $I'$ be denoted by $\bar{I}_{\bar{q},I'}$. Further, by definition of $\bar{I}^2_{\bar{q},I}$, all intervals in it occur as unions of those in $\bar{I}^2_{\bar{p},I}$, and so does $I'$. Let $\bar{I}_{\bar{p},I'} \subseteq \bar{I}^2_{\bar{p},I}$ be these intervals whose union gives $I'$. Repeat the same argument as above to obtain

$$\begin{aligned}
\|\hat{f}_{\bar{I}^2_{\bar{q}}} - f\|_{I'} &\leq \|\hat{f}_{\bar{I}_{\bar{p}}} - f\|_{\bar{I}^1_{\bar{p},I'}} + \alpha \sum_{p \in \bar{p}^1_{I'}} r_d \cdot \epsilon \sqrt{(d+1)p} \\
&\quad + \|\hat{f}_{\bar{I}^2_{\bar{q},I'}} - f\|_{\bar{I}^2_{\bar{q},I'}} + \alpha \sum_{q \in \bar{q}^2_{I'}} r_d \cdot \epsilon \sqrt{(d+1)q}, \quad (10)
\end{aligned}$$

where each of $\bar{I}^1_{\bar{p},I'}, \bar{p}^1_{I'}, \bar{I}^2_{\bar{q},I'}$ and $\bar{q}^2_{I'}$ are defined in exactly the same manner as was for $I$, but by replacing $I'$ in all definitions. Since $\bar{I}^1_{\bar{p},I'} \subseteq \bar{I}^2_{\bar{p},I} \subseteq \bar{I}_{\bar{p}}$, substituting Equation (10) into (9), a larger portion of $I$ is bounded using the difference $\|\hat{f}_{\bar{I}_{\bar{p}}} - f\|$. Upon repeating the same argument for all $\|\hat{f}_{\bar{I}_{\bar{q}}} - f\|$ terms that remain, a bound on the RHS is obtained that consists exclusively of $\|\hat{f}_{\bar{I}_{\bar{p}}} - f\|$. The entire procedure is shown in Figure 7.

Further, from Lemma 14, the sum of all the $\epsilon$-deviation terms that results on the RHS from repeating the argument is bounded by $\sqrt{2}/(\sqrt{2}-1)$ times the total $\epsilon$-deviation in $\bar{I}^3_{\bar{p}}$. This results in

$$\begin{aligned}
\|\hat{f}_I - f\|_I &\leq \|\hat{f}_{\bar{I}_{\bar{p}}} - f\|_I + \frac{\alpha}{\sqrt{2}-1} \sum_{p \in \bar{p}_I} r_d \cdot \epsilon \sqrt{(d+1)p} \\
&\stackrel{(a)}{\leq} (r_d + 1) \cdot \|f^\star_{\bar{I}_{\bar{p}}} - f\|_I + \left(1 + \frac{\alpha\sqrt{2}}{\sqrt{2}-1}\right) \sum_{p \in \bar{p}_I} r_d \cdot \epsilon \sqrt{(d+1)p},
\end{aligned}$$

where $(a)$ follows from Theorem 1. Repeating across $I \in \bar{I}_{\bar{q}}^3$ gives

$$\|\hat{f}_{\bar{I}_{\bar{q}}} - f\|_{\bar{I}_{\bar{q}}^3} \leq (r_d + 1) \cdot \|f_{\bar{I}_{\bar{p}}}^{\star} - f\|_{\bar{I}_{\bar{q}}^3} + \left(1 + \frac{\alpha\sqrt{2}}{\sqrt{2}-1}\right) \sum_{I \in \bar{I}_{\bar{p}}^3} r_d \cdot \epsilon\sqrt{(d+1)p_I}.$$

**Lemma 14.** *Suppose in the run of* MERGE, *a collection of consecutive intervals $\bar{I}_1$ was merged in $k-1$ steps to generate $\bar{I}_k$, and suppose $\bar{I}_2, \ldots, \bar{I}_{k-1}$ are the intermediate interval collections. Then,*

$$\sum_{i=1}^{k} \sum_{I \in \bar{I}_i} \sqrt{q_I} \leq \sum_{I \in \bar{I}_1} \frac{\sqrt{2}}{\sqrt{2}-1} \sqrt{q_I}.$$

**Proof** WLOG assume $\bar{q}_{\bar{I}_1}$ is a distribution, which also implies $\bar{q}_{\bar{I}_i}$ is a distribution $\forall i \in \{2, \ldots, k\}$. Notice that for any $i \in \{2, \ldots, k\}$, $\bar{q}_{\bar{I}_i} \in \Delta_{\text{bin,n}, \geq \bar{q}_{\bar{I}_{i-1}}}$. Thus $|\bar{I}_i| \leq 1/2 \cdot |\bar{I}_{i-1}|$, where $|\bar{I}|$ denotes the number of intervals in $\bar{I}$. By concavity of $\sqrt{x}$ for $x \geq 0$, the sum $\sum_{I \in \bar{I}_i} \sqrt{q_I}$ is maximized for a given $\bar{q}_{\bar{I}_{i-1}}$, if $|\bar{I}_{i-1}| = 2 \cdot |\bar{I}_i|$. Since this equality is attained iff $\bar{q}_{\bar{I}_{i-1}}$ is the uniform distribution over $|\bar{I}_{i-1}|$ elements and $\bar{q}_{\bar{I}_i}$ is uniform over $|\bar{I}_i| = 1/2 \cdot |\bar{I}_{i-1}|$ elements,

$$\sum_{I \in \bar{I}_i} \sqrt{q_I} \leq \frac{1}{\sqrt{2}} \sum_{I \in \bar{I}_{i-1}} \sqrt{q_I}.$$

This implies

$$\sum_{i=1}^{k} \sum_{I \in \bar{I}_i} \sqrt{q_I} \leq \sum_{i=1}^{k} \left(\frac{1}{\sqrt{2}}\right)^{i-1} \cdot \sum_{I \in \bar{I}_1} \sqrt{q_I} \leq \frac{\sqrt{2}}{\sqrt{2}-1} \sum_{I \in \bar{I}_1} \sqrt{q_I}.$$

# E Additional Lemmas

**Lemma 15.** *Adapting [2] to achieve a factor-2 approximation for $\mathcal{P}_2$ results in $\epsilon_n = \tilde{\mathcal{O}}(n^{-1/4})$.*

**Proof** WLOG fix the interval be $[0, 1]$. Further if $h \in \mathcal{P}_2$ has an $\ell_1$ norm $> 2$, it follows that for any distribution $f$, $\|h - f\|_1 \geq \|h\|_1 - \|f\|_1 \geq 1$. Thus WLOG restrict the $\mathcal{P}_2$ to the subset $\mathcal{Q} \stackrel{\text{def}}{=} \{h \in \mathcal{P}_2 : \|h\|_1 \leq 2, h \geq 0\}$

Let $\mathcal{D}_\epsilon$ be an arbitrary $\epsilon/2$ cover of $\mathcal{Q}$. Thus for the estimate $\hat{f}_{\text{BK}}$ output by [2],

$$\mathbb{E}\|\hat{f}_{\text{BK}} - f\|_1 = 2\text{OPT}_{\mathcal{D}_\epsilon}(f) + \tilde{\mathcal{O}}\left(|\mathcal{D}_\epsilon|^{1/5}/n^{2/5}\right) \leq 2\text{OPT}_{\mathcal{P}_2}(f) + \epsilon + \tilde{\mathcal{O}}\left(|\mathcal{D}_\epsilon|^{1/5}/n^{2/5}\right). \quad (11)$$

For $\bar{c} = (c_0, c_1) \in \mathbb{R}^2$, let $h_{\bar{c}} \stackrel{\text{def}}{=} c_0 + c_1 x + c_2 x^2$. Consider $\mathcal{C}_\epsilon$, a subset of $\mathcal{Q}$ defined as $\mathcal{C}_\epsilon \stackrel{\text{def}}{=} \{h_{\bar{c}} \in \mathcal{Q} : c_i = \lambda_i \epsilon, \ \lambda_i \in \mathbb{Z}, \ i \in \{0, 1, 2\}\}$. It is easy to see that $|\mathcal{C}_\epsilon| \geq \Omega(1/\epsilon^3)$. Since the $\ell_1$ norm between any two members in $\mathcal{C}_\epsilon$ is at least $\epsilon/2$, $|\mathcal{D}_\epsilon| \geq |\mathcal{C}_\epsilon| \geq \Omega(1/\epsilon^3)$.

Optimizing Equation (11) w.r.t. $\epsilon$ results in

$$\mathbb{E}\|\hat{f}_{\text{BK}} - f\|_1 \leq 2\text{OPT}_{\mathcal{P}_1}(f) + \tilde{\mathcal{O}}(n^{-1/4}).$$

**Lemma 16.** *Let $f$ be a Gaussian distribution. Then for a constant d, $OPT_{\mathcal{P}_{t,d}}(f) = \mathcal{O}(1/t^{d-1})$.*

**Proof** Let $t = t_1 - 2$. WLOG assume $f$ has mean 0 and variance 1 so that $f = 1/\sqrt{2\pi} \cdot e^{-x^2/2}$. Fix an $L > 0$. Divide $[-L, L]$ into $t$ equal sized intervals of length $l \stackrel{\text{def}}{=} 2L/t$. Let $h_d$ be the $t$-piecewise order $d$ Taylor polynomial of $f$ on that interval. Then for any $x \in \mathbb{R}$, $|h_d(x) - f(x)| \leq f^{d+1}(c)l^{d+1}/(d+1)!$, where $0 \leq c \leq l$.

Since $f^{d+1}(x) = H_{d+1}(x)e^{-x^2/2}$ where $H_{d+1}$ is the $d+1$th-order Hermite polynomial, standard bounds [11] imply that there exists a constant $c_d : f^{d+1}(x) \leq c_d, \forall x \in \mathbb{R}$.

Observe that on the interval, $(L, \infty)$, from standard sub-gaussian inequalities,

$$\int_L^\infty \frac{1}{\sqrt{2\pi}} e^{-x^2/2} dx \le e^{-L^2/2}.$$

Similarly for $(-\infty, L)$, extend $h_d$ to these intervals to obtain a $t + 2$-piecewise polynomial. Then

$$\|h_d - f\|_1 \le t \cdot c_d \frac{l^{d+1}}{(d+1)!} + e^{-L^2/2} = c_d \frac{(2L)^{d+1}}{t^d(d+1)!} + e^{-L^2/2}$$

Choosing $L = \mathcal{O}(\sqrt{2d \log t})$ gives $\|h_d - f\|_1 \le \mathcal{O}\left(\left((\log t)^{\frac{d+1}{2}} + 1\right)/t^d\right) = \mathcal{O}(1/t^{(d-1)})$.

# F    MERGE and COMP Algorithms

This section provides a detailed description of MERGE and COMP, the main routines of SURF. We also restate the necessary definitions.

## F.1    The MERGE Routine

MERGE receives as input, $X^{n-1}$ and parameters $d, \alpha, \epsilon$. The routine operates in $i \in \{1, \dots, D\}$ steps. Define $D(i) \overset{\text{def}}{=} D - i$ and let

$$\bar{u}_i \overset{\text{def}}{=} \left(1/2^{D(i)}, \dots, 1/2^{D(i)}\right), \quad \bar{I}_{\bar{u}_i} = (I_{\bar{u}_i, 1}, \dots, I_{\bar{u}_i, 2^{D(i)}}).$$

Initialize $\bar{q}_0 \leftarrow (1/n, \dots, 1/n)$.

Start with $i = 1$ and assign $\bar{s} \leftarrow \bar{q}_{i-1}$. In each step, the routine maintains this $\bar{s} = \bar{q}_{i-1} \in \Delta_{\text{bin,n},\le \bar{u}_i}$. This can be seen from the initialization above for $i = 1$ since $\bar{u}_1 = (2/n, \dots, 2/n)$, and verified for $i > 1$. Thus, using $\bar{I}_{\bar{u}_i}$, we may separate

$$\bar{I}_{\bar{s}} = (\bar{I}_{\bar{s},1}, \dots, \bar{I}_{\bar{s}, 2^{D(i)}}), \quad \bar{s} = (\bar{s}_1, \dots, \bar{s}_{2^{D(i)}}),$$

where for each $j \in \{1, \dots, 2^{D(i)}\}$, $\bar{I}_{\bar{s},j} \subseteq \bar{I}_{\bar{s}}$ are intervals in $\bar{I}_{\bar{s}}$ whose union gives $I_{\bar{u}_i, j} \in \bar{I}_{\bar{u}_i}$. Let $\bar{s}_j \in \bar{s}$ denote the empirical probabilities in $\bar{s}$ corresponding to intervals in $\bar{I}_{\bar{s},j}$. Notice that the sum of all probabilities in $\bar{s}_j$, $\sum_{s \in \bar{s}_j} s = 1/2^{D(i)}$. Therefore the scaled $2^{D(i)} \bar{s}_j$ is an empirical distribution. For brevity, let the polynomial estimate output by INT on $I_{\bar{u}_i, j}$, be denoted by

$$\hat{f}_{I_j} \overset{\text{def}}{=} \hat{f}_{I_{\bar{u}_i, j}, \text{INT}}.$$

Starting with $j = 1$, invoke COMP with arguments, the polynomial estimate $\hat{f}_{I_j}$, intervals $\bar{I}_{\bar{s},j}$ and the empirical distribution $2^{D(i)} \bar{s}_j$, samples $X_{i,j}^{n-1} \subseteq X^{n-1}$ that lie in $I_{\bar{s},j}$, and parameters $d$,

$$\gamma \overset{\text{def}}{=} \alpha \cdot r_d \cdot \epsilon \sqrt{d+1}.$$

This parameter, $\gamma$, is used to tune the bias-variance trade-off. As will be shown subsequently, if $\gamma \to \infty$, $\bar{I}_{\bar{s},j}$ will be merged, resulting in an estimate with a larger bias but smaller variance. A small $\gamma$ has the opposite effect.

If $\text{COMP}(\hat{f}_{I_j}, \bar{I}_{\bar{s},j}, 2^{D(i)} \bar{s}_j, X_{i,j}^{n-1}, d, \gamma) \le 0$, merge $\bar{I}_{\bar{s},j}$ into a single interval $I_{\bar{u}_i, j}$. Accomplish this by updating $\bar{s}_j$ to a unitary value, its sum, $(1/2^{D(i)})$. Otherwise, maintain $\bar{s}$ as is. Increment $j$ within the range $\{1, \dots, 2^{D(i)}\}$ and repeat this procedure.

After the entire run in $j$ is complete, update $\bar{q}_i \leftarrow \bar{s}$. If $D(i) = D - i > 0$, increment $i$ and repeat the same steps. Otherwise, if $D(i) = 0$ or in other words if $i = D$, MERGE, and in turn, SURF outputs the piecewise estimate on $\bar{I}_{\bar{q}_D}$, i.e. $\hat{f}_{\text{SURF}} = \hat{f}_{\bar{I}_{\bar{q}_D}, \text{INT}}$.

At each step $i \in \{1, \dots, D\}$, MERGE calls COMP on $2^{D(i)}$ intervals, each consisting of $2^i$ samples. Thus each step of MERGE takes $\mathcal{O}(2^{D(i)} \cdot (d^\tau + \log(2^i)) \cdot 2^i) = \mathcal{O}((d^\tau + \log n)2^D)$ time. The total time complexity is therefore $\mathcal{O}((d^\tau + \log n)2^D D) = \mathcal{O}((d^\tau + \log n)n \log n)$.

---

**Algorithm 1** MERGE

---

**Input:** $X^{n-1}, d, \alpha, \epsilon$
Initialize $D = \log n, \bar{q} = (1/n, \ldots, 1/n), \gamma \leftarrow \alpha \cdot r_d \epsilon \cdot \sqrt{d+1}$
**for** $i = 1$ **to** $D$ **do**
   $D(i) \leftarrow D - i, \bar{s} \leftarrow \bar{q}$
   **for** $j = 1$ **to** $2^{D(i)}$ **do**
      **if** $\mathrm{COMP}(\hat{f}_{I_j}, \bar{I}_{\bar{s},j}, 2^{D(i)}\bar{s}_j, X_{i,j}^{n-1}, d, \gamma) \leq 0$ **then**
         $\bar{s}_j \leftarrow (1/2^{D(i)})$
      **end if**
   **end for**
   $\bar{q} \leftarrow \bar{s}$
**end for**
**Output:** $\bar{q}$

---

### F.2 The COMP Routine

COMP receives as input, a function $\hat{f}$, an interval partition $\bar{I} \stackrel{\text{def}}{=} \bar{I}_{\bar{s}}$ and the corresponding empirical distribution $\bar{s}$, samples $X^m$ that lie in $\bar{I}$, and parameters $d, \gamma$.

Fix a $\bar{p} \in \Delta_{\mathrm{bin},m,\geq\bar{s}}$, and consider the piecewise polynomial estimate on $\bar{I}_{\bar{p}}$, $\hat{f}_{\bar{I}_{\bar{p}},\mathrm{INT}}$. Define

$$\Lambda_{\bar{I}_{\bar{p}}}(\hat{f}) \stackrel{\text{def}}{=} \|\hat{f}_{\bar{I}_{\bar{p}},\mathrm{INT}} - \hat{f}\|_{\bar{I}_{\bar{p}}}, \ \lambda_{\bar{p},\gamma} \stackrel{\text{def}}{=} \sum_{p\in\bar{p}} \gamma\sqrt{p}. \tag{12}$$

$\mathrm{COMP}(\hat{f})$ returns $\mu_{\bar{I}_{\bar{s}},\gamma}(\hat{f})$, the largest difference between $\Lambda_{I_{\bar{p}}}(\hat{f})$ and $\lambda_{\bar{p},\gamma}$ across all $\bar{p} \in \Delta_{\mathrm{bin},m,\geq\bar{s}}$,

$$\mu_{\bar{I}_{\bar{s}},\gamma}(\hat{f}) \stackrel{\text{def}}{=} \max_{\bar{p}\in\Delta_{\mathrm{bin},m,\geq\bar{s}}} \Lambda_{\bar{I}_{\bar{p}}}(\hat{f}) - \lambda_{\bar{p},\gamma}.$$

The quantity, $\Lambda_{\bar{I}_{\bar{p}}}(\hat{f})$ acts as a proxy for the increment in bias that results if the piecewise estimate $\hat{f}_{\bar{I}_{\bar{p}},\mathrm{INT}}$ is merged into $\hat{f}$, while $\lambda_{\bar{p},\gamma}$ accounts for the deviation in $\hat{f}_{\bar{I}_{\bar{p}},\mathrm{INT}}$ under $\mathcal{Q}_\epsilon$. Notice that for any $\bar{p} \in \Delta_{\mathrm{bin},m,\geq\bar{s}}$, $\lambda_{\bar{p},\gamma} \leq \lambda_{\bar{s},\gamma}$. Thus $\mu_{\bar{I}_{\bar{s}},\gamma}(\hat{f}) \leq 0$ if the decrease in deviation under $\bar{I} = \bar{I}_{\bar{s}}$ is larger than the increased bias under any candidate $\bar{I}_{\bar{p}}$. This in turn signals MERGE to merge $\bar{I}$.

It may be shown that if $\bar{s} = (1/m, \ldots, 1/m)$, the cardinality, $|\Delta_{\mathrm{bin},m,\geq\bar{s}}| = \Omega(m^c)$ for any $c > 0$. Therefore, naively evaluating $\Lambda_{I_{\bar{p}}}(\hat{f}) - \lambda_{\bar{p},\gamma}$ over each $\bar{p} \in \Delta_{\mathrm{bin},m,\geq\bar{s}}$ incurs a worst case time complexity that is super-linear in $m$. Instead, COMP uses a simple divide-and-conquer procedure that computes $\mu_{\bar{I}_{\bar{s}},\gamma}(\hat{f})$ in time $\mathcal{O}((d^\tau + \log m)m)$.

To describe this, notice that if $\bar{I}_{\bar{s}}$ is a singleton $(I)$, then $\bar{s} = (1)$, implying $\Delta_{\mathrm{bin},m,\geq\bar{s}} = \{(1)\}$. In this case, obtain $\hat{f}_{I,\mathrm{INT}} \in \mathcal{P}_d$ and return

$$\mu_{\bar{I}_{\bar{s}},\gamma}(\hat{f}) = \Lambda_{\bar{I}_{(1)}}(\hat{f}) - \lambda_{(1),\gamma} = \|\hat{f}_{I,\mathrm{INT}} - \hat{f}\|_{\bar{I}_{(1)}} - \gamma\sqrt{1}.$$

If $\bar{I}_{\bar{s}}$ is non singleton or $\bar{s} \neq (1)$, any $\bar{p} \in \Delta_{\mathrm{bin},m,\geq\bar{s}} \setminus \{(1)\}$ may be split into two sub-distributions, $\bar{p}_1, \bar{p}_2$ that each sum to $1/2$. For example, if the particular $\bar{p} = (1/4, 1/4, 1/8, 1/8, 1/4)$, it may be split into $\bar{p}_1 = (1/4, 1/4)$ and $\bar{p}_2 = (1/8, 1/8, 1/4)$. The corresponding interval partition is also split into $\bar{I}_{\bar{p}} = (\bar{I}_{\bar{p}_1}, \bar{I}_{\bar{p}_2})$. Since $\bar{s} \neq (1)$, this may also be similarly split into $\bar{s}_1$ and $\bar{s}_2$. As a consequence, $\bar{I}_{\bar{s}}$ is also cleaved into $(\bar{I}_{\bar{s}_1}, \bar{I}_{\bar{s}_2})$ corresponding to $\bar{s}_1$ and $\bar{s}_2$. Using this observation,

$$\max_{\bar{p}\in\Delta_{\mathrm{bin},m,\geq\bar{s}},\, \bar{p}\neq(1)} \Lambda_{\bar{I}_{\bar{p}}}(\hat{f}) - \lambda_{\bar{p},\gamma} = \max_{\bar{p}\in\Delta_{\mathrm{bin},m,\geq\bar{s}},\, \bar{p}\neq(1)} \Lambda_{\bar{I}_{\bar{p}_1}}(\hat{f}) - \lambda_{\bar{p}_1,\gamma} + \Lambda_{\bar{I}_{\bar{p}_2}}(\hat{f}) - \lambda_{\bar{p}_2,\gamma}$$

$$= \max_{\bar{p}_1\in\Delta_{\mathrm{bin},m/2,\geq2\bar{s}_1}} \Lambda_{\bar{I}_{\bar{p}_1}}(\hat{f}) - \lambda_{\bar{p}_1,\gamma/\sqrt{2}}$$

$$+ \max_{\bar{p}_2\in\Delta_{\mathrm{bin},m/2,\geq2\bar{s}_2}} \Lambda_{I_{\bar{p}_2}}(\hat{f}) - \lambda_{\bar{p}_2,\gamma/\sqrt{2}}$$

$$= \mu_{\bar{I}_{2\bar{s}_1},\gamma/\sqrt{2}}(\hat{f}) + \mu_{\bar{I}_{2\bar{s}_2},\gamma/\sqrt{2}}(\hat{f}),$$

**Algorithm 2** COMP

> **Input:** $\hat{f}, \bar{I}_{\bar{s}}, \bar{s}, X^m, d, \gamma$
> $I \leftarrow \cup \bar{I}_{\bar{s}}, \; \mu \leftarrow \Lambda_I(\hat{f}) - \lambda_{\bar{s},\gamma}$
> **if** $|\bar{I}| = 1$ **then**
>     **Return:** $\mu$
> **else**
>     **Return:** $\max\{\mu, \text{COMP}(\hat{f}, \bar{I}_{\bar{s}_1}, 2\bar{s}_1, X_1^m, d, \gamma/\sqrt{2})$
>         $+ \text{COMP}(\hat{f}, \bar{I}_{\bar{s}_2}, 2\bar{s}_2, X_2^m, d, \gamma/\sqrt{2})\}$
> **end if**

where $2\bar{s}_1, 2\bar{s}_2$ are the normalized variants of $\bar{s}_1, \bar{s}_2$, and $\gamma$ is scaled by $1/\sqrt{2}$ to accommodate for this scaling. By evaluating $\mu_{\bar{I}_{2\bar{s}_1}, \gamma/\sqrt{2}}(\hat{f}), \mu_{\bar{I}_{2\bar{s}_1}, \gamma/\sqrt{2}}(\hat{f})$ separately, and then comparing their sum with $\Lambda_{\bar{I}_{(1)}}(\hat{f}) - \lambda_{(1),\gamma}$, we allow for a recursive computation of $\mu_{\bar{I}_{\bar{s}}, \gamma}(\hat{f})$.

Let $X_1^m$ and $X_2^m$ denote the samples in $\bar{I}_{\bar{s}_1}$ and $\bar{I}_{\bar{s}_2}$ respectively. Using these arguments, call COMP on $\bar{I}_{\bar{s}_1}, \bar{s}_1$ and $\bar{I}_{\bar{s}_2}, \bar{s}_2$, return the maximum as shown in Algorithm 2.

Now $\Lambda_{\bar{I}_{(1)}}(\hat{f}) - \lambda_{(1),\gamma}$ is calculated by obtaining $\hat{f}_{I,\text{INT}} \in \mathcal{P}_d$ from INT. Since $I$ has $m$ samples, from Theorem 1, this takes $\mathcal{O}(m + d^\tau)$ time. Further, notice that since both $\bar{s}_1$ and $\bar{s}_2$ sum to $1/2$, the split $\bar{I}_{\bar{s}} = (\bar{I}_{\bar{s}_1}, \bar{I}_{\bar{s}_2})$ occurs along the median of $X^m$. Thus $\bar{I}_{\bar{s}_1}$ and $\bar{I}_{\bar{s}_2}$ has at most half the number of samples, $m/2$, and the time complexity of COMP, $T(m)$, is captured by

$$T(m) \leq 2T(m/2) + \mathcal{O}(m + d^\tau),$$

implying $T(m) = \mathcal{O}((d^\tau + \log m)m)$.

### F.3 Distributed Computation of COMP and MERGE

We consider the scenario where we are provided with pre-sorted samples, a known $t$ and $\Theta(m)$ memory for some $t \leq m \leq n$. Let a unit of memory be equivalent to that which is required to store the value of one sample. In this case, we may split the available memory to simulate $m$ concurrent processors with constant processing memory. WLOG let $0 \leq t \leq n$ and for simplicity, let $t, m$ be a power of 2, just like $n$. Define $D_t \overset{\text{def}}{=} \log_2 t, D_m \overset{\text{def}}{=} \log_2 m$ and recall that $D \overset{\text{def}}{=} \log_2 n$.

Let $\text{MERGE}_t$ be the modified MERGE that halts in $D - D_t$ steps instead of $D$. The corresponding $\text{SURF}_t$ outputs the polynomial estimate corresponding to the interval partition given by $\bar{q}_{D-D_t}$ (instead of the one corresponding to $\bar{q}_D$ output by SURF). Let this estimate be denoted by $\hat{f}_{\text{SURF}_t} \overset{\text{def}}{=} \hat{f}_{\text{INT}, I_{\bar{q}_{D-D_t}}}$ and let $\bar{u}_t \overset{\text{def}}{=} (1/2^{D_t}, \ldots, 1/2^{D_t}) = (1/t, \ldots, 1/t)$ be the uniform distribution on $t$ intervals.

**Lemma 17.** *Given samples $X^{n-1} \sim f$, for some $t < n$ that are both powers of 2, degree $d \leq 8$ and the threshold $\alpha > 2$, $\text{SURF}_t$ outputs $\hat{f}_{\text{SURF}_t}$ in time $\mathcal{O}((d^\tau + \log n)n \log n)$ such that under event $\mathcal{Q}_\epsilon$,*

$$\|\hat{f}_{\text{SURF}_t} - f\|_1 \leq \min_{\bar{p} \in \Delta_{\text{bin,n}}, \leq \bar{u}_t} \sum_{I \in \bar{I}_{\bar{p}}} \left( \frac{(r_d + 1)\alpha}{\alpha - 2} \inf_{h \in \mathcal{P}_d} \|h - f\|_I \right.$$

$$\left. + \frac{r_d(\alpha\sqrt{2} + \sqrt{2} - 1)}{\sqrt{2} - 1} \epsilon \sqrt{(d+1)q_I} \right),$$

*where $q_I$ is the empirical mass under interval $I$, $r_d$ is the constant in Theorem 1.*

As argued in Theorem 2, Lemma 17 along with Lemma 9 implies that $\hat{f}_{\text{SURF}_t}$ is an $r_d$-factor approximation for $\mathcal{P}_{t,d}$.

For $1 \leq i \leq D$, recall that $D(i) \overset{\text{def}}{=} D - i$. In step $i$ of $\text{MERGE}_t$, COMP is called on sub-intervals $\bar{I}_{\bar{s},j} \subseteq \bar{I}_{\bar{s}}$ for $j \in \{1, \ldots, 2^{D(i)}\}$, generated by the interval partition $\bar{I}_{(1/2^{D(i)}, \ldots, 1/2^{D(i)})}$. Each $\bar{I}_{\bar{s},j}$, for $j \in \{1, \ldots, 2^{D(i)}\}$ consists of $n/2^{D(i)} = 2^i$ samples.

| (a) Densities $f_1, f_2$. | (b) $\ell_1$ error with $d = 1$. | (c) $\ell_1$ error with $d = 2$. | (d) $\ell_1$ error with $d = 3$. |

Figure 8: Evaluation of the estimate output by SURF with degrees $d = 1, 2, 3$, $\alpha = 0.25$, on $f_1 = 0.3\mathcal{N}(0.4, 0.1^2) + 0.7\mathcal{N}(0.6, 0.2^2)$ and $f_2 = 0.4\mathcal{N}(0.3, 0.05^2) + 0.6\mathcal{N}(0.7, 0.15^2)$.

| (a) Densities $f_1, f_2$. | (b) $\ell_1$ error with $d = 1$. | (c) $\ell_1$ error with $d = 2$. | (d) $\ell_1$ error with $d = 3$. |

Figure 9: Evaluation of the estimate output by SURF with degrees $d = 1, 2, 3$, $\alpha = 0.25$, on $f_1 = 0.2\text{Gam}(4, 0.04) + 0.8\text{Gam}(8, .06)$ and $f_2 = 0.4\text{Gam}(3, 0.05) + 0.6\text{Gam}(6, .075)$.

Given presorted samples, for the steps $i$ such that $2^{D(i)} \leq m$, each call to COMP may be implemented concurrently on the $m$ processors. This results in a time complexity of $\mathcal{O}((d^\tau + \log n)2^i)$ for that step, where $\tau \in [2, 2.4)$ is the matrix inversion constant. As $2^{D(i)} \leq m$ implies $D(i) = D - i \leq \log_2 m = D_m$, or $i \geq D - D_m$,. The total time taken by these steps is thus given by $\sum_{i=D-D_m}^{D-D_t} \mathcal{O}((d^\tau + \log n)2^i) = \mathcal{O}((d^\tau + \log n)2^{D-D_t}) = \mathcal{O}((d^\tau + \log n)n/t)$.

For steps $i$ in the range $1 \leq i < D - D_m$, COMP may be implemented concurrently in batches, with each batch consisting of $m$ sub-intervals among $\bar{I}_{\bar{s},1}, \ldots, \bar{I}_{\bar{s},j}$. As there are a total of $2^{D(i)}/m$ batches, and as each interval consists of $2^i$ samples, step $i$ takes time $\mathcal{O}((d^\tau + \log n)2^i) \cdot 2^{D(i)}/m$ $= \mathcal{O}((d^\tau + \log n)n/m)$. The total time taken by steps $1 \leq i < D - D_m$ is given by $\mathcal{O}((d^\tau + \log n)n/m) \cdot (D - D_m) = \mathcal{O}((d^\tau + \log n)n \log n/m)$.

Thus the time complexity under distributed computation is $\mathcal{O}((d^\tau + \log n)n \max\{1/t, \log n/m\})$.

## G   Additional Experiments

This section shows additional experiments on the Gaussian and gamma mixtures. Just as in Section 5, SURF is run with $\alpha = 0.25$ and the results are averaged over 10 runs.

Since SURF is invariant to location-scale transformations, WLOG we run experiments on distributions such that essentially all its mass lies in the interval $[0, 1]$. Let $\mathcal{N}(\mu, \sigma^2)$ be the Gaussian distribution with parameters $\mu$, $\sigma$. We run SURF with degrees $d = 1, 2, 3$ on the two Gaussian mixtures shown in Figure 8(a). Figures 8(b)–8(d) show the resulting $\ell_1$ errors. This is repeated for the gamma mixture density shown in Figure 9, where $\text{Gam}(k, \theta)$ denotes the gamma distribution with shape, scale parameters $k$, $\theta$ respectively. Figures 9(b)–9(d) show the corresponding $\ell_1$ errors.

Notice that the errors are similar between distributions, and that the error saturates more quickly for $d = 3$, as the higher degree allows SURF to exploit the smoothness inherent in the considered parametric families. These observations are in line with what was observed for the beta mixtures considered in Figure 2.