[Reviews · NeurIPS 2020]

Review 1

Summary and Contributions: The article focuses on density estimation in $\mathbb R$. The authors develop an adaptive estimator based on an local polynomials estimator. In particular, the authors improve the results of "Sample-optimal density estimation in nearly-linear time" in several aspects: - They improve the constant factor in front of the bias term when the degree of polynomials is smaller than $d=8$. - They improve the time complexity. - They obtain an adaptive procedure allowing to choose the number of local polynomials (which is the main advantage of this procedure). - The computations can be distributed. The theoretical results are accompanied with interesting simulations. UPDATE: I would like to thank the authors for their feedback. However, it does not change my evaluation of the paper.

Strengths: The strenghts of their procedure is both practical an theoretical: - Theory: (See contributions) - Practice: The computations may be distributed The main advantage is that the procedure is adaptive with respect to the number of local polynomials, which is not the case of "Sample-optimal density estimation in nearly-linear time". Consequently the result can be used in the same manner when the density is either log-concave or either a mixture of Poisson distributions (for examples) .

Weaknesses: Theoretically, it could be possible to use the results from "Sample-optimal density estimation in nearly-linear time" and a cross validation to obtain an adaptive procedure with respect to the number of local polynomials (this is claimed by the authors). It leads to a factor 15 in front of the bias term. However their residual term would be smaller than the one presented in this article. Consequently, when the bias is small, the advantage of the procedure is limited. Since local polynomials have a good representation power, in practice, the bias may be smaller than the residual term, limiting the usefulness of the paper. That's why you should other examples examples in the paper. You present only the problem of learning Gaussiant densities.

Correctness: I have a little concern about Corollary 3. In the proof of Corollary 3 you choose $\alpha \rightarrow \infty$. Consequently, Corollary 3 is wrong. It does not holds for any $\alpha \geq 2$. Moreover, something is unclear for me. Such a choice of $\alpha$ leads to $\gamma \rightarrow \infty$ and thus a choice of always merging ... There is something weird, here don't you think ? It's because for large $\alpha$ the residual term explodes. There is a trade-off to precise !

Clarity: Here is a latex code for typos and remark I have on this point. \section{Typos} \begin{enumerate} \item Line 36: "Since $f \in \mathcal C$", a "for" is missing. \item line 110: $\mathcal P_{t,f}$ should be $\mathcal P_{f}$ I think. \item Line 205: twice "operates" \item Proof Corollary 3: You write $\hat f_{\mathcal A(X^{(n-1)})}$ instead of $\hat f_{SURF}$. \end{enumerate} \section{Comments} \begin{enumerate} \item Is the distribution $f$ continuous ? In this case, the $\ell_1$ norm equals the TV norm (with a factor 2). Otherwise, the distance between a discrete and continuous probability measure is not well defined. You should write the definition with TV or restrict yourself to continuous distributions. Moreover, with TV definition, it becomes clear why $\| u- v \| =2$ in line 29. \item We do not say "min-max" but "minimax rates". \item Could you please introduce the t-piecewise extension $\mathcal P_{t,f}$. \item In Theorem 2 and Corollary 3 we use the parameter $\alpha$ which is not introduced clearly. Since you present $\gamma$ in the sequel (which depends on $\alpha$, you should write everything with $\gamma$. \item The notation $\Delta_{\mathbb R}$ should be replaced by $\Delta_{\mathbb R}(X^n)$ to show the dependence with respect to the sample $(X_1,\cdots,X_n)$. \item Part 2 could be simplified. Honestly, it's hard to follow you ... For example, you define "interval probabilities" and "empirical probabilities" and then use "interval empirical probabilities. Please work on that ...For example, you don't introduce $\Delta$ in (1). It took me 15 min to understand that actually: \[ \Delta_{emp,n} = \big \{ (a_1/n,\cdots,a_k/n), a_i \in \mathbb N^* \textnormal{ s.t } \sum a_i = n, k \in \{1,\cdots,n\} \big \} \] Also I think the definition if $\bar I_{\bar q}$ is wrong for the first interval. \item Line 158: in my opinion the notation $I_{\bar n_{d},i}$ is misleading because too close to $I_{a,b}$. \item Definition $\Delta_{bin,n}$ should be more precise (same problem as $\Delta_{emp,n}$). \item Is Lemma 9 hold for any $d \geq 1$ ? Could you precise it in the statement please. \item The "COMP" procedure actually uses the standard "bias-variance" trade-off in non-parametric statistics (see the book "Introduction to nonparametric estimation"). This is something that should be explained. The "COMP" procedure looks complicated at a first sight, but is quite common actually. \item In my opinion you should put the array comparing SURF and ADLS in the introduction. It will be simpler for the reader to grasp the advantage of your procedure. \item In the proof of Corollary 3 you choose $\alpha \rightarrow \infty$. Consequently, Corollary 3 is wrong. It does not holds for any $\alpha \geq 2$. Moreover, something is unclear for me. Such a choice of $\alpha$ leads to $\gamma \rightarrow \infty$ and thus a choice of always merging ... There is something weird, here don't you think ? \item You insist on the fact the "SURF" is robust. I don't understand in which sense it is robust ? Robustness has a precise sense in the statistical literature. Could you precise it please. \end{enumerate}

Relation to Prior Work: Density estimation is a very common problem in the statistical community. The literature here is poor and could be largely improved. You should also use the vocabulary from this community. For example, you approach is based on the approach used in "Consistency of data-driven histogram methods for density estimation and classification". The goal is to develop a data-driven way of selecting a partition of $\mathbb R$ and fit local polynomials of degree $d$ in each interval. The choice of the partition depends on the COMP procedure which is based on the well-known bias-variance trade-off.

Reproducibility: Yes

Additional Feedback: The article is interesting, the ideas are nice but the article is poorly written. It could be easily improved (see some of my comments). I am aware that the limited number of pages prevents from precising everything, but some notations and arguments could be more precise. I think you should rewrite the article and submit it in a journal. It will be much clearer for the reader. The article is hard to follow, but if you rewrite it and precise many statements it could be a very good article.


Review 2

Summary and Contributions: This paper develops an algorithm for density estimation of structured one-dimensional probability distributions with respect to the \ell_1-distance. More specifically, an efficient "agnostic" algorithm for piecewise polynomial densities is developed. While an efficient algorithm for the same setting had been developed in prior work, the proposed algorithm adapts better to unknown number of interval pieces and gives a somewhat better approximation ratio when the degree of the polynomial is small. A basic experimental evaluation and comparison is performed.

Strengths: At the theoretical level, an interesting contribution is the fact that the proposed algorithm is adaptive with respect to the unknown number of intervals. Moreover, when the degree of the polynomial is small (at most 8) the constant factor approximation achieved is <3, as opposed to 3 in the most relevant prior work ADLS. These properties seem to yield some experimental improvements for synthetic datasets

Weaknesses: A limitation of the current work is that any potential theoretical/practical improvements over ADLS require the assumption that the degree d<8. The authors argue that this ought to hold in practice, though I found this justification somewhat weak. For example, it is not clear how the approximation ratio r_d behaves for d>=8. The experimental evaluation is not sufficiently comprehensive. Given that one of the main comparison approaches (in terms of performance) is the prior work ADLS, a more detailed comparison would have been important for real datasets as well, not only for a few synthetic ones.

Correctness: The theoretical results appear to be correct. The experimental methodology appears to be sound.

Clarity: The paper is generally clearly written, modulo some comments/questions provided below.

Relation to Prior Work: The comparison to prior work is overall appropriate. Some specific points regarding the relation to the prior work ADLS and the experiments were somewhat confusing and would benefit from clarification.

Reproducibility: Yes

Additional Feedback: Some questions for the authors: 1) With respect to the table on line 251. ** How does r_d behave for d>=8? What is the limit when d increase to infinity? ** With respect to the time complexity: According to the SODA'17 version of ADLS, the running time of their algorithm is O~(n) for all degrees d, as long as they relax the sample complexity by a logarithmic factor. (See remark after the main theorem statement in that paper.) As a result, the runtime comparison in that table could be misleading to the reader. 2) In the real data experiments, it was not immediately clear what the extent of the improvement is over various kernel density estimators. Moreover, a comparison to ADLS is not provided in these graphs. ** Thanks to the authors for addressing these questions. Overall, my evaluation remains the same. The paper is somewhat borderline from a theory point of view. The experimental evaluation is nice, but not sufficiently convincing for NeurIPS.


Review 3

Summary and Contributions: The paper presents an algorithm for distribution estimation using piecewise polynomial functions. The authors explain the algorithm and analyze its runtime and sample complexities. The authors compare previous similar algorithms theoretically and empirically and show that their algorithm achieves better performance in time and accuracy for low-degree polynomials.

Strengths: The task of distribution estimation is an essential problem in machine learning. Improving current algorithms is good progress in the ability to handle this task. The paper shows both theoretical and analytical improvements, which strengthen the confidence in its ability to generalize. The analysis requires almost no assumptions on the input data. The paper is well written and provides good intuition of the work and the claims.

Weaknesses: - The “related work” subsection (1.2) feels poor and is lack of information about other approaches rather than ADLS. There might be more room to elaborate on the given citations. Mainly, the following information is missing: Is there an algorithm that gives a better approximation/runtime? Why not also comparing to spline and MLE-based methods that can also provide polynomial estimations? - The algorithm estimates distributions using piecewise polynomials. The theoretical guarantee is an improvement over previous works only for low degree polynomials (at most 8). However, it is not rare that theoretical results are more conservative than empirical ones. Therefore, It would be worthwhile to see in practice how the algorithm behaves for higher degrees, comparing to other algorithms. - As for the generalization results, how tight is theorem 2 are the theory bounds in practice? It would be interesting to add them to the experimental results.

Correctness: While I did not thoroughly review the proofs, the proofs' intuitions do not raise any red flag.

Clarity: The paper is well written and organized.

Relation to Prior Work: The authors provide great emphasis to explain their contribution over ADLS method. However, it is lack of information about other approaches (see weaknesses for more details).

Reproducibility: Yes

Additional Feedback:

[Meta-Review · NeurIPS 2020]

The paper develops a simple approximation method for one-dimensional densities using piecewise polynomials, which adapts to the number of pieces. While all the reviewers are borderline positive, they bring up a few issues regarding both the theory and the experiments (and their general opinion has not changed after the rebuttal).